# On the Connection Between Counterfactually fair representation, Statistical Parity and Individual Fairness

## Abstract

The relations among observational fairness notions (those defined based on data distributions) have been studied in the literature, yet the relations between counterfactual fairness and observational fairness notions remain less explored. In this paper, we study the relations between counterfactual fairness and two kinds of observational fairness, statistical parity and individual fairness. In particular, we are interested in understanding whether a predictor trained using counterfactually fair representations (CFR)(Zuo et al., 2023) can satisfy individual fairness and statistical parity. We show that, for a certain type of causal model called the Gaussian Causal Model (GCM), counterfactual fairness through CFR can imply both statistical parity and individual fairness. We also identify another class of causal models under which counterfactual fairness through CFR implies statistical parity. Experiments on both synthetic and real-world data demonstrate that counterfactually fair representation can enhance fairness in machine learning models without compromising performance, outperforming methods designed for observational fairness.

## 1 Introduction

As machine learning (ML) models play an increasingly integral role in modern society, there is a growing public concern about the potential risks associated with these models (Abadie & Kasy, 2017). Thus, in addition to high performance, it is crucial to ensure the trustworthiness of ML models. One of the key aspects of building trustworthy ML models is ensuring fairness. Unfortunately, it has been well evidenced that ML models may exhibit biases against certain social groups. For example, the COMPAS recidivism prediction tool was found to exhibit bias against African Americans (Dieterich et al., 2016). Even state-of-the-art language models, such as ChatGPT, have demonstrated encoded stereotypes about gender (Gross, 2023). Addressing these fairness concerns is crucial for having a trustworthy ML model.

To tackle unfairness issues in ML, various fairness notions have been proposed, including 1) *fairness through unawareness* (Fabris et al., 2023), which defines fairness by prohibiting models from using sensitive attributes; 2) *parity-based fairness* such as statistical parity (Besse et al., 2022), equal opportunity (Wang et al., 2019; Hardt et al., 2016), equalized odds (Romano et al., 2020), which requires certain statistical measures to be equalized across different groups; 3) *preference-based fairness*, which draws inspiration from fair-division and envy-freeness in economics, ensures every group would favor its own decision outcomes than decisions of any other groups (Zafar et al., 2017); 4) *individual fairness*, which indicates that individuals having similar attributes should also receive similar prediction/decision.

Unlike the above fairness notions that are defined based on observational data, counterfactual fairness (Kusner et al., 2017) is another type of fairness notion that is defined based on an underlying causal model. In particular, within the causal framework, counterfactual fairness creates a hypothetical counterfactual world and requires the distribution of the predicted label for an individual in the factual world to remain the same as that in the counterfactual world where the individual belongs to another social group. In other words, counterfactual fairness requires an individual to be treated equally in the factual and counterfactual worlds.

In general, there is an inherent tension between different notions of fairness, and satisfying one notion can contradict others (Friedler et al., 2021). But one may wonder whether there exist such scenarios under which

different fairness notions may be compatible and can hold simultaneously. Indeed, the connections between different fairness notions have been explored in the literature. For example, prior works have shown that only under highly constrained special cases, parity-based fairness such as statistical parity, equal opportunity, and predictive rate parity can be compatible (Kleinberg et al., 2016; Chouldechova, 2017). However, the connections between counterfactual fairness and observational fairness are relatively unexplored. To the best of our knowledge, only a few recent works (Anthis & Veitch, 2023) studied the relations between causal fairness and statistical parity. However, they are limited to very constrained causal models where no exogenous variables are included. This is because counterfactual reasoning in causal models with no exogenous variables is easier and more straightforward. Victor et al. (2021) and Fawkes & Evans (2023) also consider a similar causal model for finding a counterfactual invariant predictor (Quinzan et al., 2022), and their results are valid where there are no exogenous variables. There have been efforts to make a connection between counterfactual fairness and statistical parity (Rosenblatt & Witter, 2023). However, Silva (2024) argues that the results in (Rosenblatt & Witter, 2023) do not always hold. When the dataset generated by the causal model exhibits a selection bias, the selection can also affect the relationship between counterfactual fairness and other fairness notions (Fawkes et al., 2022). Apart from counterfactual invariant, there are several other causality based fairness notions (Makhlouf et al., 2020) including causal predictive parity (Plecko & Bareinboim, 2024) and counterfactual principal fairness (Nilforoshan et al., 2022). In this paper, we focus our discussion on the counterfactual fairness notion provided by Kusner et al. (2017).

In this work, we rigorously examine the connection between counterfactual fairness (CF), statistical parity (SP), and individual fairness (IF). In particular, we are interested in understanding whether a predictor trained using counterfactually fair representations (Zuo et al., 2023), which is an extension to the algorithm proposed in Kusner et al. (2017), can satisfy individual fairness and statistical parity simultaneously. We will identify conditions on causal models under which *counterfactual fairness yields statistical parity and individual fairness*. Using both real and synthetic data, we compare the performance of predictors trained on counterfactually fair representations with predictors trained by existing methods for statistical parity or individual fairness. We will show that predictors trained on counterfactually fair representation can satisfy multiple fairness notions simultaneously without significant accuracy drop compared to the baselines.

The rest of the paper is organized as follows. We present the preliminaries in Section 2, followed by our theoretical results in Section 3. We display the experiments on synthetic data and real world datasets in Section 4, and conclude in Section 5.

## 2 Preliminaries

### 2.1 Definition of counterfactual fairness, statistical parity and individual fairness

We consider a supervised learning problem with a training dataset consisting of triplets $(A, X, Y)$, where $A \in \mathcal{A}$ is a sensitive attribute distinguishing individuals from multiple groups (e.g., race, gender), $X \in \mathcal{X}$ is a feature vector, and $Y \in \mathcal{Y}$ is the target/label. The goal is to learn a predictor from training data that can predict $Y$ given inputs $A$ and $X$. Let $\hat{Y} = g_y(X, A)$ denotes the output of the predictor $g_y$ given an input $(X, A)$.

In this work, we want to study the relation between counterfactual fairness, individual fairness, and statistical parity. To do that, first, we provide the definition of counterfactual fairness, individual fairness, and statistical parity. Counterfactual fairness is defined based on a Structural Causal Model (SCM) (Pearl, 2010). A SCM is denoted by $\mathcal{M}(U, V, F)$ and consists of three sets: a set of exogenous (unobservable) variables $U$, a set of endogenous (observable) variables $V$, and a set of structural equations $F$. $V$ is the union of the sensitive attribute $A$, the feature vector $X$ and the target attribute $Y$. An element $f_i$ in $F$ determines the causal relationship between an observed feature $V_i \in V$ and its parent attributes $U_{pa_i} \subseteq U$ and $V_{pa_i} \subseteq V$. That is[1],

$$V_i = f_i(U_{pa_i}, V_{pa_i}). \tag{1}$$

---

[1]The structural equation can be relaxed into conditional distributions when the causal relationship is not deterministic. In this case, $V_i \sim \mathcal{D}(U_{pa_i}, V_{pa_i}; v_i)$. $\mathcal{D}(U_{pa_i}, V_{pa_i}; \cdot)$ is a distribution parameterized by $U_{pa_i}$ and $V_{pa_i}$.

In SCM, exogenous variables $U$ are associated with a prior distribution $P(U)$. The structural equations $F$ enable us to perform counterfactual inference and calculate **counterfactual quantities**. Counterfactual inference enables us to answer the following question: What would be the value/distribution of an observable variable $Z$ if $Q \in V$ had taken value $q$. Since the distribution of any observable variable is determined by unobserved variables $U$ and causal relationships, the counterfactual value of $Z$ given $U = u$ can be computed by replacing $U$ with the value $u$ in structural equations and replacing structural equation for $Q$ by $Q = q$. The resulting counterfactual value for $Z$ is denoted by $Z_{Q \leftarrow q}(u)$. Further, if we are interested in calculating counterfactuals given an observation $O = o$ ($O$ is a set of observable variables), we can take advantage of SCMs to calculate $\Pr\{Z_{Q \leftarrow q}(U) = z | O = o\}$. This probability helps us to find "the distribution of $Z$ if $Q$ had taken value $q$ in the presence of evidence $O = o$". Distribution $\Pr\{Z_{Q \leftarrow q}(U) = z | O = o\}$ can be calculated in the following steps: 1. *abduction:* we find the posterior distribution of $U$ given $O = o$; (ii) *action:* we apply an intervention $Q = q$ by replacing the structural equation of $q$ with $Q = q$; (iii) *prediction:* we compute the distribution of $Z$ by new structural equations and the posterior distribution $\Pr\{U = u | O = o\}$.

Given the above preliminaries, the definition of counterfactual fairness (Kusner et al., 2017) for $\hat{Y}$ is as follows,

$$\Pr\{\hat{Y}_{A \leftarrow a}(U) = y | X = x, A = a\} = \Pr\{\hat{Y}_{A \leftarrow \check{a}}(U) = y | X = x, A = a\} \quad \forall y \in \mathcal{Y}, X \in \mathcal{X}, a, \check{a} \in \mathcal{A},$$

The above definition implies that for an individual with $(X = x, A = a)$, the prediction $\hat{Y}$ in the factual world should have the same marginal distribution as that in the counterfactual world in which the individual belongs to a different group.

Statistical Parity (SP) fairness notion and individual fairness (IF) notion, on the other hand, are solely defined based on the joint probability distribution function of $X, \hat{Y}$ and $A$. SP requires $A$ and $\hat{Y}$ to be independent (i.e., $\Pr\{\hat{Y} = y | A = a\} = \Pr\{\hat{Y} = y\}, \forall y \in \mathcal{Y}, \forall a \in \mathcal{A}$). Individual fairness ensures that two individuals with similar characteristics or attributes receive similar treatments/decisions. Mathematically, it can be defined as follows (Dwork et al., 2012),

$$d_1\left(g_y(x, a), g_y(x', a')\right) \leq d_2\left((x, a), (x', a')\right), \tag{2}$$

where $d_1$ and $d_2$ are two distance functions.

## 2.2 Counterfactually fair representation

In order to satisfy counterfactual fairness, we follow the method proposed in Zuo et al. (2023), which is an extension of the algorithm in Kusner et al. (2017), and to our best knowledge, it is the most general way to guarantee counterfactual fairness. In particular, we generate a counterfactually fair representation associated with each input $(x, a)$, and train a predictor using such a representation to satisfy counterfactual fairness. The counterfactually fair representation is defined as follows.

**Definition 1** (Counterfactually Fair Representation (CFR)). Consider a structural causal model $\mathcal{M}(U, V, F)$ where $V = (A, X, Y)$, a representation $H(X, A)$ associated to $X$ and $A$ is counterfactually fair if

$$\Pr\{H_{A \leftarrow a}(U) = h | X = x, A = a\} = \Pr\{H_{A \leftarrow \check{a}}(U) = h | X = x, A = a\} \quad \forall h \in \mathcal{H}, X \in \mathcal{X}, a, \check{a} \in \mathcal{A},$$

where $\mathcal{H}$ is the representation space.

In this work, we focus on generating a counterfactually fair representation $H$ associated with each data point $(x, a)$, and train a predictor using such a representation. We follow the algorithm proposed in Zuo et al. (2023) to generate such a representation. It has been shown that any predictor whose input is counterfactually fair representation satisfies counterfactual fairness (Zuo et al., 2023). To construct a counterfactually fair representation proposed by Zuo et al. (2023), we first need to generate counterfactual samples defined as follows.

**Definition 2** (Counterfactual Sample). Consider a data point $(x, a)$, and let $U$ be the unobservable variable associated with $(x, a)$ sampled from the distribution $\Pr_{\mathcal{M}}\{U | X = x, A = a\}$ under causal model $\mathcal{M} = (V, U, F)$ (subscript $\mathcal{M}$ implies that the probability is calculated based on the causal model $\mathcal{M}$). Then, $(\check{x}, \check{a})$

is a counterfactual sample with respect to $(x, a)$ if it is generated using structural equations $F$, unobservable variable $U = u$, and intervention $A = \check{a}$.[2] We denote the random variable associated with $\check{x}[\check{a}]$ by $\check{X}[\check{a}]$. In particular, $\check{X}[\check{a}]$ is a random variable generated based on causal model $\mathcal{M}$ with intervention $A = \check{a}$ and $U$ following posterior distribution $\Pr_{\mathcal{M}}\{U|X = x, A = a\}$. Sample $\check{x}[\check{a}]$ is the realization of random variable $\check{X}[\check{a}]$.

Next, we introduce how to generate a counterfactually fair representation for a given data point $(X = x, A = a)$ in general. The first step is to infer the conditional distribution of $U$ given the data point $(X = x, A = a)$. The representation can be constructed in two ways:

- Consider a symmetric function $s$, which means the order of inputs will not affect its output. A counterfactually fair representation can be defined as $H(x, a) = [s\left(\mathbb{E}\left[\check{X}[\check{a}^{[1]}]|U\right], ..., \mathbb{E}\left[\check{X}[\check{a}^{|\mathcal{A}|}]|U\right]\right), U]$[3], where $U$ follows the conditional distribution $\Pr_{\mathcal{M}}\{U|X = x, A = a\}$.[4] The representation $H(x, a)$ is still a random variable which is a function $U$, and the distribution of $U$ follows $\Pr_{\mathcal{M}}\{U|X = x, A = a\}$. Based on Zuo et al. (2023), to use $H(x, a)$ as the input of machine learning models, we use a realization of $H(x, a)$ denoted by $h(x, a) = [s\left(\mathbb{E}\left[\check{X}[\check{a}^{[1]}]|U = u\right], ..., \mathbb{E}\left[\check{X}[\check{a}^{|\mathcal{A}|}]|U = u\right]\right), u]$ where $u$ is sampled from $U|X = x, A = a$. Note that, $\mathcal{A} = \{a^{[1]}, a^{[2]}, \ldots, a^{[|\mathcal{A}|]}\}$, and $a^{[i]} \neq a^{[j]}$.

- The second way of constructing the representation is to calculate the following expectation, $r(x, a) = \mathbb{E}_{U \sim \Pr_{\mathcal{M}}\{U|X=x, A=a\}}\left[s(\mathbb{E}\left[\check{X}[\check{a}^{[1]}]|U\right], ..., \mathbb{E}\left[\check{X}[\check{a}^{[|\mathcal{A}|]}]|U\right]), U\right]$. Note that $r(x, a)$ is a deterministic representation, not a random variable.

To illustrate the construction process of the two kinds of representation, we provide a simple example in the following.

**Example 1.** Consider a SCM $M(U, V, F)$ shown in Figure 1. When $X = x, A = a$ is given, an instantiation $h(x, a)$ of the counterfactually fair representation $H(x, a)$ is built as the following steps:

1. Find the posterior distribution $\Pr_{\mathcal{M}}(U|X = x, A = a)$ and sample $u$ from the distribution;

2. Compute $x = f(u, a)$ and $\check{x} = f(u, \check{a})$. If $f$ is a randomized function, we compute $x = \mathbb{E}[f(u, a)]$ and $\check{x} = \mathbb{E}[f(u, \check{a})]$ where the expectation is over the randomness of $f$;

3. The representation $h(x, a) = [s(x, \check{x}), u]$.

Counterfactual fair representation $r(x, a)$ has the same first two steps, but in the last step, $r(x, a) = \mathbb{E}_{U \sim \Pr_{\mathcal{M}}(U|X=x, A=a)}[s(x, \check{x}), u]$. It should be noticed that because $r(x, a)$ is related to the distribution of $U$, it can not be written as a function of $U$.

Note that in practice, in order to find the output of a machine learning model for $H(x, a)$, we find a realization of $H(x, a)$ and then calculate the output using the realization (as explained in Zuo et al. (2023)). While this process satisfies counterfactual fairness, it would cause a large variance in output. On the other hand, $r(x, a)$ is a deterministic representation, and the output associated with $r(x, a)$ has zero variance. So, using $r$ may be more desirable if we want to avoid variance, while the discussion of $r$ was neglected in Romano et al. (2020)

## 2.3 Problem formulation

---

[2] Counterfatual sample can be generated for any $\check{a} \in \mathcal{A}$.

[3] $|\mathcal{A}|$ is the size of the domain, so $a^{[1]}, a^{[2]}, ..., a^{|\mathcal{A}|}$ represents all possible values of $A$

[4] For calculating $\mathbb{E}\left[\check{X}[\check{a}]|U\right]$, we can first calculate $\mathbb{E}\left[\check{X}[\check{a}]|U = u\right]$ which is a function of $u$ denoted by $e(u)$. The randomness in this expectation comes from the causal model and structural equations. Then, $\mathbb{E}\left[\check{X}[\check{a}]|U\right] = e(U)$.

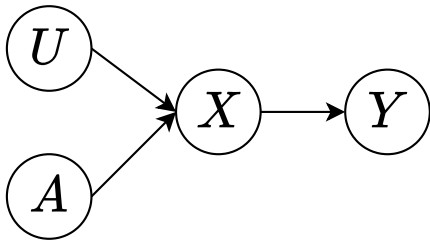

Figure 1: An example SCM for CFR construction illustration

In order to satisfy counterfactual fairness, we follow the method proposed in Zuo et al. (2023), which is an extension of the algorithm in Kusner et al. (2017), and to our best knowledge, it is the most general way to guarantee counterfactual fairness. In particular, we generate a counterfactually fair representation associated with each input $(x, a)$, and train a predictor using such a representation to satisfy counterfactual fairness. Our goal is to answer the following questions:

1. Does a predictor trained using the counterfactually fair representation satisfy statistical parity? If the answer is yes, then we can take advantage of counterfactually fair representations to achieve both counterfactual fairness and statistical parity.

2. Does a predictor trained using the counterfactually fair representation satisfy individual fairness with respect to $(x, a)$ when the predictor is individually fair for the representation. If the answer is yes, then we can take advantage of counterfactually fair representations to achieve individual fairness as well.

We try to answer the above questions in this paper. In particular, we identify conditions under which the answers to the above questions are yes. As a result, the findings in this paper show there is a possibility to achieve all of these fairness notions at the same time.

## 3 Theoretical Result

### 3.1 Connection between CF and IF

In this section, we introduce a special kind of causal model, called Gaussian Causal Model (GCM). The structural functions in GCM are non-deterministic.[5]

**Definition 3** (Gaussian Causal Model (GCM)). A structural causal model $\mathcal{M}(U, V, F)$ is a Gaussian causal model (GCM) if the following conditions hold:

1. $P(U)$ is a Gaussian distribution

$$U \sim \text{Gaussian}(\mu_u, \Sigma_u^2), \tag{3}$$

2. Structural functions for $X$ is given by,

$$X \sim \text{Gaussian}(W_u U + f_a(A) + b, \Sigma_x^2), \tag{4}$$

where $f_a$ is an arbitrary function[6].

---

[5]Counterfactual fairness under non-deterministic structural functions has also been studied in Kusner et al. (2017).

[6]Suppose $U$ is a $d$-dimensional vector, $[WU]_i$ is the value in the $i$-th dimension of the vector $WU$, $WU + f_a(A)$ is calculated in a broadcasting way, i.e. $[WU + f_a(A)]_i = [WU]_i + f_a(A)$.

With the definition of GCM, we have the following theorems which reveal the connection between counterfactual fairness and individual fairness.

**Theorem 1.** Given a Gaussian causal model, if $A \perp U$, and $f_a$ is Lipschitz continuous[7], then the counterfactually fair representation $H(x, a)$ satisfies the following,

$$d\left(H(x, a), H(x', a')\right) \leq L_1 ||(x, a) - (x', a')||_2 \quad \forall x, x', a, a'. \tag{5}$$

If we denote $\bar{s}(u) = s\left(\mathbb{E}\left[\check{X}[\check{a}^{[1]}]|U = u\right], ..., \mathbb{E}\left[\check{X}[\check{a}^{[|\mathcal{A}|]}]|U = u\right]\right)$ and $\bar{s}(u)$ is Lipschitz continuous, then we have,

$$\|r(x, a) - r(x', a')\|_2 \leq L_2 ||(x, a) - (x', a')||_2 \quad \forall x, x', a, a', \tag{6}$$

where $L_1, L_2$ are constants determined by the causal model, the Lipschitz constants of $f_a$ and $\bar{s}$. Moreover, $d(.,.)$ is the total variation (Takezawa, 2005) measuring the distance between two distributions, and $\|\cdot\|_2$ is the Euclidean norm.

Theorem 1 shows that the representations $H(x, a)$ and $r(x, a)$ are Lipschitz continuous. The proof for this theorem can be seen in the Appendix. Next, we show that if $H(x, a)$ and $r(x, a)$ are Lipschitz continuous, the model trained on them can satisfy individual fairness with respect to $(x, a)$.

**Theorem 2.** Assume that Equations 5 and 6 hold for representations $H(x, a)$ and $r(x, a)$. Then, for any predictor $g$, we have,

$$d\left(g(H(x, a)), g(H(x', a'))\right) \leq L_1 \|(x, a) - (x', a')\|_2 \quad \forall x, x', a, a', \tag{7}$$

where $d$ is the total variation of two distributions. If $g$ is Lipschitz continuous with a Lipschitz constant $L_g$, we have,

$$\|g(r(x, a)) - g(r(x', a'))\|_2 \leq L_g L_2 ||(x, a) - (x', a')||_2 \quad \forall x, x', a, a'. \tag{8}$$

Theorem 1 and 2 together show that under certain conditions, counterfactual fairness can imply individual fairness.[8] On the other hand, in GCMs, individual fairness does not necessarily imply counterfactual fairness under the same conditions.. The following example shows that even under the Gaussian causal model, we can find an optimal predictor that satisfies individual fairness while it is counterfactually unfair.

**Example 2.** Consider a Gaussian causal model with $f_a(a) = W_a a$ (where $W_a$ is a vector with the same size of $X$). The target $Y \sim \text{Gaussian}(W_x X + b_x, \Sigma_x^2)$. The predictor $\hat{Y}(X, A) = W_x W_u \mathbb{E}_{U \sim \text{Pr}_{\mathcal{M}}\{U|X, A\}}[U] + W_x W_a A + W_x b + b_x$ is optimal and satisfies individual fairness since $\left\|\hat{Y}(x, a) - \hat{Y}(x', a')\right\|_2 \leq \sqrt{2} \|W_x W_u C\|_2 (\max\{1, \|W_a\|_2\} + \|C^{-1} W_u^{-1} W_a\|_2) \|(x, a) - (x', a')\|_2$[9]. However, given $x, a$, we have $\text{Pr}\{\hat{Y}_{A \leftarrow a}(U) = \hat{y}|X = x, A = a\} = \delta\left(\hat{y} - W_x W_u \mathbb{E}_{U \sim \text{Pr}_{\mathcal{M}}\{U|X=x, A=a\}}[U] - W_x W_a a - W_x b - b_x\right)$, while $\text{Pr}\{\hat{Y}_{A \leftarrow \check{a}}(U) = \hat{y}|X = x, A = a\} = \delta\left(\hat{y} - W_x W_u \mathbb{E}_{U \sim \text{Pr}_{\mathcal{M}}\{U|X=x, A=a\}}[U] - W_x W_a \check{a} - W_x b - b_x\right)$. That means the predictor will give different prediction in the factual and counterfactual world. Hence, counterfactual fairness is violated.[10]

### 3.2 Connection between CF and SP under GCM

Statistical Parity (SP) is a group fairness criterion that implies the prediction should be independent of the sensitive attribute. We can prove that under the Gaussian causal model, a counterfactually fair representation can result in SP.

---

[7]The domain of $A$ could be $\mathcal{A} = \{a^{[1]}, a^{[2]}, ..., a^{[|\mathcal{A}|]}\}$ or a continuous range $\mathcal{A} = [a, b] \subset R$. Lipschitz continuous of $f_a$ means $\forall a, a' \in \mathcal{A}, |f_a(a) - f_a(a')| \leq L_a |a - a'|$.

[8]Similar results hold when $U$ follows a Gamma distribution. See Appendix C for more details.

[9]$C = (W_u^{\mathrm{T}} \Sigma_x^{-1} W_u + \Sigma_u^{-1})^{-1} W_u^{\mathrm{T}} \Sigma_x^{-1}$

[10]A step-by-step explanation of this example can be seen in Appendix E

**Theorem 3.** Consider a Gaussian Causal Model. If $A \perp U$, then counterfactually fair representation $H(X, A)$ satisfies $H(X, A) \perp A$. Moreover, if $A \perp U$, and $f_a(a)$ is linear, i.e. $f_a(a) = W_a a$, and $A$ follows a uniform distribution $(\Pr\{A = a\} = \Pr\{A = a'\}, \forall a, a' \in \mathcal{A})$, $r(X, A) \perp A$.

Note that if $H(X, A)$ and $r(X, A)$ are independent of $A$, $g(H(X, A))$ and $g(r(X, A))$ are independent of $A$ as well, where $g$ is a deterministic function/predictor. Therefore, by Theorem 3, under certain conditions, a predictor with the counterfactually fair representation as its input satisfies statistical parity.

### 3.3 Connection between CF and SP beyond GCM

In this part, we want to focus on a kind of causal model other than GCM. In particular, we consider structural equations $X_i = f_i(U_{pa_i}, V_{pa_i})$ with $f_i$ being a deterministic function. In this type of causal model, if $A$ does not have any parent, we can substitute every $X_j$ in $V_{pa_i}$ by $f_j(U_{pa_j}, V_{pa_j})$ iteratively to write $X_i$ as a function of $U$ and $A$. We denote the mapping from $U$ and $A$ to $X_1, ..., X_n$ by $f$. In particular, we can write $X = f(U, A)$. In this type of causal model, we can show that under certain conditions, counterfactual fairness implies statistical parity.

**Theorem 4.** Consider a causal model $\mathcal{M}(U, V, F)$ with $X = f(U, A)$, and counterfactually fair representation $H(X, A)$ or $r(X, A)$, If $A \perp U$, we have $H(X, A) \perp A$. Moreover, if the following conditions hold,

1. $A \perp U$,

2. $P(U)$ is a uniform distribution $(\Pr\{U = u\} = \Pr\{U = u'\}, \forall u, u' \in \mathcal{U})$,

3. For any $a, a'$ and $u, u'$, $f(u, a) = f(u', a) \Leftrightarrow f(u, a') = f(u', a')$,

$r(X, A) \perp A$, i.e. counterfactual fairness implies statistical parity.

This theorem shows that any machine learning model $g$ trained on the counterfactually fair representation $H(X, A)$ or $r(X, A)$ has the same output distribution across different groups and satisfies SP when the underlying causal model satisfies the conditions.

In general, under conditions of Theorem 4, statistical parity does not imply counterfactual fairness. To show this, we provide the following example.

**Example 3.** We construct a causal model $\mathcal{M}$ that satisfies the conditions in Theorem 4 and a representation $R$ which is independent of $A$ but does not satisfy counterfactual fairness. Consider a causal model that consists of one exogenous variable $U$, a sensitive attribute $A$, and observed variables $X \in \mathbb{R}$ and $Y \in \mathbb{R}$. The prior distribution of $U$ is a uniform distribution defined over $[-1, 1]$. Random variable $A$ follows the Bernoulli distribution,

$$P(A = 1) = 0.5, \ P(A = -1) = 0.5. \tag{9}$$

So we have the first two conditions satisfied. Then, we consider the following structural functions, $X = U \cdot A$, $Y = X$. Since this is a bijective model, the third condition in Theorem 4 is satisfied. We can construct a representation $R = X$ and the prediction function $\hat{Y} = R$. The predictor is optimal in the sense of mean squared error. We have,

$$P(R = r|A = 1) = P(U = r), \tag{10}$$

$$P(R = r|A = -1) = P(U = -r). \tag{11}$$

Therefore, we have $P(R = r|A = 1) = P(R = r|A = -1)$ implying $R \perp A$. Next consider a sample $X = x, A = a$. It is easy to see that $\Pr\{R_{A \leftarrow a} = r|X = x, A = a\} = 1$ if $r = x$. On the other hand, $\Pr\{R_{A \leftarrow (-a)} = r|X = x, A = a\} = 1$ if $r = -x$. This shows that $\Pr\{R_{A \leftarrow a} = r|X = x, A = a\} \neq \Pr\{R_{A \leftarrow (-a)} = r|X = x, A = a\}$, and $R$ is not counterfactually fair.

Theorem 3 and Theorem 4 imply that using counterfactually fair representations $H(X, A)$ or $r(X, A)$ is a viable way to achieve CF and SP simultaneously, but Example 3 reminds that when a prediction satisfies SP, it might not be counterfactually fair.

We want to emphasize that Rosenblatt & Witter (2023) also tries to study the relation between counterfactual fairness and statistical parity. However, in Rosenblatt & Witter (2023), the authors only consider a case where the prediction is a function of $U$, the proof techniques of which might be able to applied to the case of using $H(x, a)$. However, when the counterfactual fair prediction is not a function of $U$, for example, a predictor using $r(x, a)$ as the input, the conditions they found to show that counterfactual fairness implies statistical parity is not enough. Example 4 in the Appendix provides a specific counter-example to show the results in Rosenblatt & Witter (2023) is not applicable to our setting. In Silva (2024), the author also points out that counterfactual fairness and statistical parity are not generally the same. Our results are consistent with Silva (2024) as we show that under assumptions $A \perp U$, counterfactual fairness and statistical parity can not imply each other. However, they do not provide conditions when the two fairness are related. Further, we show under stronger assumptions, there exists a scenario where counterfactual fairness can imply statistical parity.

## 4 Experiment

### 4.1 Experiment with synthetic data generated by GCM

In this section, we generate $N = 10000$ data according to an GCM. In the simulation, $U$ and $X$ are 5-dimensional vectors. $U$ are sampled from a standard normal distribution (i.e., normal distribution with zero mean and identity covariance matrix). $A$ is binary and set to 1 or 2 with equal probabilities. We generat the target variable $Y$ with a linear function of $X$. The other parameters in GCM can be found in the Appendix. To show whether the counterfactually fair representation (CFR) can achieve statistical parity and individual fairness at the same time, we compare it with two baselines. The first baseline is the unfair linear regression model (UF) trained without any fairness constraints. The second baseline is GLIF (Petersen et al., 2021) which post-processes the prediction of the unfair predictor to satisfy individual fairness.[11] For the counterfactually fair method, we used $r(X, A) = \mathbb{E}_{U \sim \mathrm{Pr}_{\mathcal{M}}(U|X,A)}[U]$[12] as the representation. Then we used a linear regression model to take $r(X, A)$ as input.

Consider the dataset $\{a^{(i)}, x^{(i)}, y^{(i)}\}_{i=1}^N$, we denote the prediction for the $i$-th data by $\hat{y}^{(i)}$. We use mean squared error (MSE) to evaluate the predictive performance, which is defined as $\mathrm{MSE} = \frac{1}{N} \sum_{i=1}^N (y^{(i)} - \hat{y}^{(i)})^2$. To evaluate whether the predictor satisfies statistical parity, we train an SVM classifier with $(\hat{y}^{(i)}, a^{(i)})$ to predict $a^{(i)}$ using $\hat{y}^{(i)}$. We denote the output of SVM by $\hat{a}^{(i)}$ and calculate A-Accuracy as follows, $\mathrm{A-Accuracy} = \frac{1}{N} \sum_{i=1}^N \mathbf{1}(a^{(i)}, \hat{a}^{(i)})$, where $\mathbf{1}(a^{(i)}, \hat{a}^{(i)}) = 1$ if $a^{(i)} = \hat{a}^{(i)}$, otherwise is 0. If a predictor satisfies SP, the A-Accuracy should be 50%. For individual fairness, we use IF-ratio defined as follows, $\mathrm{IF-Ratio} = \frac{1}{N^2} \sum_{i=1}^N \sum_{j=1, j\neq i}^N \frac{\left\| \hat{y}^{(i)} - \hat{y}^{(j)} \right\|_2}{\left\| (x^{(i)}, a^{(i)}) - (x^{(j)}, a^{(j)}) \right\|_2}$. For counterfactual fairness, we use total effect (TE). Assume the prediction on the counterfactual data $(\check{a}^{(i)}, \check{x}^{(i)})$ is $\check{y}^{(i)}$, TE is defined as $\mathrm{TE} = \frac{1}{N} \sum_{i=1}^N \left| \hat{y}^{(i)} - \check{y}^{(i)} \right|$. We split the generated dataset into train/test sets with a ratio of 80%/ 20% randomly 5 times and compute the average metrics. The results are displayed in Table 1. As we can see in this table, a model trained by the counterfactually fair representations can achieve both statistical parity and individual fairness at the same time. Compared to the GLIF method, even though we satisfy the individual fairness with a larger IF-Ratio constant, the MSE is much smaller. We also notice that the GLIF method does not satisfy statistical parity and counterfactual fairness.

---

[11]The iFair(Lahoti et al., 2019) method requires the data to be composed of several clusters, which is not suitable for the GCM generated data.

[12]We only used the expectation of $U$ here because $s\left( \mathbb{E}\left[ \check{X}[\check{a}^{[1]}]|U \right], ..., \mathbb{E}\left[ \check{X}[\check{a}^{|\mathcal{A}|}]|U \right] \right)$ did not improve the MSE on this synthetic data.

Table 1: Results on synthetic data generated by GCM. UF is the model trained without considering fairness. GLIF post-processes the output of UF model to achieve individual fairness. CFR method use the counterfactually fair representation as the input of the predictor.

| Method | MSE | A-Accuracy | IF-Ratio | TE |
|---|---|---|---|---|
| UF | $0.00 \pm 0.00$ | $62.0\% \pm 0.38\%$ | $0.581 \pm 0.005$ | $4.83 \pm 0.07$ |
| GLIF | $5.50 \pm 0.09$ | $59.9\% \pm 0.23\%$ | $0.299 \pm 0.003$ | $2.57 \pm 0.03$ |
| CFR | $2.09 \pm 0.00$ | $50.6\% \pm 0.20\%$ | $0.560 \pm 0.005$ | $0.0 \pm 0.0$ |

## 4.2 Experiment with synthetic data beyond GCM

In this section, we generate a batch of synthetic data with a known causal model to demonstrate the validity of Theorem 4. The causal graph for generating the synthetic data is shown in Figure 2a. The structural functions of the model are given by $X_0 = U_0, X_1 = b_1 + W_1^A A + W_1^X X_0 + W_1^U U, X_2 = b_2 + W_2^A A + W_2^X X_0 + W_2^U U, Y = W_Y^A A + W_Y^X X_0 + W_Y^U U$. $U$ is drawn from a uniform distribution over $[0, 1]$. $A$ is a binary attribute with $\Pr\{A = 0\} = \Pr\{A = 1\} = 0.5$. $U_0$ is sampled from the uniform distribution defined over

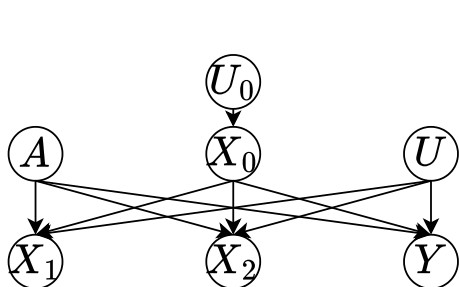

(a) Causal Graph for Generating Synthetic Data

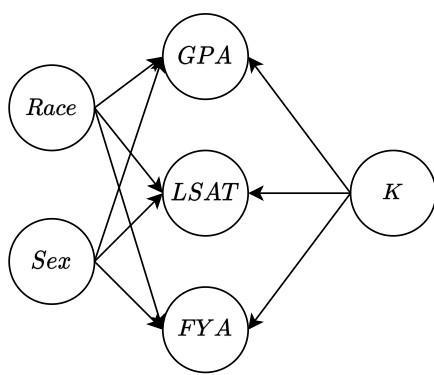

(b) Causal Graph for Law School Success Dataset

the set $\{1, ..., 8\}$ and translated into a one-hot vector. In Appendix, we provide detailed information about the data generation, including the value of the parameters, and baseline implementations. Our goal here is to validate in what extent a model trained on the counterfactually fair representation can satisfy statistical parity. Moreover, we would like to compare fairness level and accuracy with baselines for training a fair predictor under statistical parity. We denote our generated dataset by $\{x_0^{(i)}, x_1^{(i)}, x_2^{(i)}, a^{(i)}, u^{(i)}, u_0^{(i)}\}_{i=1}^N$. $N$ is the number of data instances in the dataset. Then, we generate counterfactual features $\check{x}_1$ and counterfactual features $\check{x}_2$ as $\check{x}_1^{(i)} = b_1 + W_1^A \check{a}^{(i)} + W_1^X x_0^{(i)} + W_1^U u^{(i)}$, and $\check{x}_2^{(i)} = b_2 + W_2^A \check{a}^{(i)} + W_2^X x_0^{(i)} + W_2^U u^{(i)}$. Since $U$ and $U_0$ are uniquely determined when $X = x, A = a$ is given, $h(x^{(i)}, a^{(i)})$ and $r(x^{(i)}, a^{(i)})$ are the same and the expectation can be omitted. We used $h_{CF}^{(i)}$ to denote the counterfactually fair representation, which is computed as $h_{CF}^{(i)} = \left[ u^{(i)}, x_0^{(i)}, \frac{\left(x_1^{(i)} + \check{x}_1^{(i)}\right)}{2}, \frac{\left(x_2^{(i)} + \check{x}_2^{(i)}\right)}{2} \right]$. In the same way with baselines, the fair representations $h_{CF}^{(i)}$ and target variable $y^{(i)}$ are used to train a linear regression model. We generate 30000 data and split them into a train set, validation set, and test set in the $80\%/10\%/10\%$ ratio. For every method, we randomly split the dataset 5 times and run experiments independently. We adopt four baselines and compare their performance with a predictor trained on the counterfactually fair representation. **Unfair Prediction (UF)**: The UF method is a predictor that uses the original data as the input without any fairness consideration. That is, $h_{UF}^{(i)} = [a^{(i)}, x_0^{(i)}, x_1^{(i)}, x_2^{(i)}]$ will be the input of the predictor. **CI** (Xie et al., 2017): The method consists of an encoder $E(\cdot)$, a predictor $G(\cdot)$ and a discriminator $D(\cdot)$ for training a model under SP. An encoder is used to encode the input as a representation. The predictor tries to predict the target from the representation and the discriminator tries to reveal the sensitive attribute from the

Table 2: Results on non-GCM synthetic data: comparison with 5 baselines, unfair prediction (UF), controllable-invariance (CI), maximum entropy adversarial representation learning (MaxEnt-ARL), fair representation via distributional contrastive variational autoencoder with student kernel (FarconVAE-t) and with Gaussian kernel (FarconVAE-G) in terms of performance (MSE), statistic parity (A-Accuracy) and counterfactual fairness (TE).

| Method | MSE | A-Accuracy | TE |
|---|---|---|---|
| UF | $0.00 \pm 0.00$ | $100\% \pm 0.00\%$ | $0.40 \pm 0.00$ |
| CI | $0.04 \pm 0.02$ | $64.3\% \pm 5.44\%$ | $0.18 \pm 0.06$ |
| MaxEnt-ARL | $0.02 \pm 0.01$ | $73.5\% \pm 3.47\%$ | $0.21 \pm 0.09$ |
| FarconVAE-t | $0.03 \pm 0.01$ | $71.3\% \pm 3.21\%$ | $0.23 \pm 0.05$ |
| FarconVAR-G | $0.02 \pm 0.00$ | $75.9\% \pm 0.86\%$ | $0.26 \pm 0.02$ |
| CFR | $0.04 \pm 0.00$ | $50.4\% \pm 0.28\%$ | $0.00 \pm 0.00$ |

representation. The three parts are trained adversarially to obtain the minimized prediction error and the maximized discrimination error. After training the model, $h_{CI} = E(x)$ would be the input of the predictor $G$. **MaxEnt-ARL** (Roy & Boddeti, 2019): The method is similar to CI, but when training the encoder and predictor, it uses the entropy loss instead of the discrimination loss. **FarconVAE-t** (Oh et al., 2022): The method uses a VAE structure to disentangle the latent space into sensitive-related ($h_s$) and non-sensitive-related parts ($h_x$). To train the VAE, FarconVAE-t minimizes the reconstruction loss, contrastive learning loss, and swap reconstruction loss at the same time. After training the model, for every input $x$, the encoder generates $h_x$ and $h_s$. We use $h_{FT} = h_x$ as the fair representation under statistical parity. **FarconVAE-G** (Oh et al., 2022): It is a similar method to FarconVAE-t. The difference is that FraconVAE-G uses a Gaussian-kernel for calculating the contrastive learning loss while FraconVAE-t uses a student-kernel.

Table 2 displays the results of the CFR method and baselines in terms of the three metrics. From the results, the counterfactually fair representation can achieve A-Accuracy near 50%. It means that from the counterfactually fair representation, the SVM model is unable to recover $A$. Even though we do not impose any constraints to make $A$ and $H_{CF}$ independent, $H_{CF} \perp A$ holds and the statistical parity is satisfied as we expected by Theorem 4. On the other hand, the baselines including the adversarial (CI and MaxEnt-ARL) or disentangle (FarconVAE-t and FarconVAE-G) methods cannot achieve perfect fairness in terms of SP. Moreover, all the baselines, which are designed for statistic parity, cannot achieve counterfactual fairness (which is reflected by TE). Compared to the UF baseline, the CF representation only increases MSE by a small amount to achieve both SP and CF. The increase is expected because $A$ is highly correlated with $Y$ and provides information about $Y$.

### 4.3 Experiment on the Law School Dataset

In this section, we conduct an experiment with the Law School Success dataset (Wightman, 1998). This dataset consists of 21,791 records. Each record is characterized by 4 attributes: Sex ($S_{law}$), Race ($R_{law}$), GPA ($G_{law}$) in college, LSAT ($L_{law}$), and ZFYA ($Z_{law}$) which is the first year average grade in the law school. Both Sex and Race are categorical in nature. The Sex attribute can be either male or female, while Race can be Amerindian, Asian, Black, Hispanic, Mexican, Puerto Rican, White, or Other. The GPA is a continuous variable ranging from 0 to 4. LSAT is an integer attribute with a range of [0, 60]. ZFYA, which is the target variable for prediction, is a real number ranging from -4 to 4 (it has been normalized). The goal of this dataset is to predict ZFYA from the features. In this study, we consider $S_{law}$ as the sensitive attribute, while $R_{law}, G_{law}$, and $L_{law}$ are treated as features.

The causal model for the real-world dataset is not fully known and should be constructed with the help of human knowledge. We utilize the same causal graph as Kusner et al. (2017). The causal graph is shown in Figure 2b. In this causal graph, $K_{law}$ denotes knowledge which is a hidden variable. We assume $K_{law}$

follows a uniform distribution and there is a linear relationship between the attributes, which is

$$G_{law} \sim \text{Gaussian}(b_G + W_G^S S_{law} + W_G^R R_{law} + W_G^K K_{law}, \sigma_G), \tag{12}$$

$$L_{law} \sim \text{Gaussian}(b_L + W_L^S S_{law} + W_L^R R_{law} + W_L^K K_{law}, \sigma_L), \tag{13}$$

$$Z_{law} \sim \text{Gaussian}(W_Z^S S_{law} + W_Z^R R_{law} + W_Z^K K_{law}, \sigma_G). \tag{14}$$

Unlike the synthetic data, the parameters in the causal functions remain unknown. In this experiment,

Table 3: Results on the Law School Success Dataset: comparison with 5 baselines, unfair prediction (UF), controllable-invariance (CI), maximum entropy adversarial representation learning (MaxEnt-ARL), fair representation via distributional contrastive variational autoencoder with student kernel (FarconVAE-t) and with Gaussian kernel (FarconVAE-G) in terms of performance (MSE), statistic parity (A-Accuracy) and counterfactual fairness (TE).

| Method | MSE | A-Accuracy | TE |
|---|---|---|---|
| UF | $0.75 \pm 0.03$ | $100\% \pm 0.00\%$ | $1.83 \pm 0.17$ |
| CI | $0.75 \pm 0.03$ | $57.0\% \pm 0.87\%$ | $1.71 \pm 0.40$ |
| MaxEnt-ARL | $0.76 \pm 0.03$ | $56.8\% \pm 1.18\%$ | $5.01 \pm 4.90$ |
| FarconVAE-t | $0.81 \pm 0.03$ | $62.1\% \pm 5.28\%$ | $0.71 \pm 0.40$ |
| FarconVAR-G | $0.79 \pm 0.02$ | $71.0\% \pm 15.4\%$ | $0.98 \pm 0.35$ |
| CFR | $0.79 \pm 0.02$ | $57.6\% \pm 0.98\%$ | $0.00 \pm 0.00$ |

we use the Markov Chain Monte Carlo (MCMC)(Brooks, 1998) to infer the posterior distribution of the parameters first. We sample 4000 values for each parameter and treat the mean value of the samples as the approximation. In the second stage, we use the approximate parameters to infer the distribution of $K_{law}$. The dataset can be expressed by $\{g_{law}^{(i)}, l_{law}^{(i)}, z_{law}^{(i)}, a_{law}^{(i)}, r_{law}^{(i)}\}_{i=1}^N$. For each data instance $\left(g_{law}^{(i)}, ..., r_{law}^{(i)}\right)$, we sample 4000 knowledge values $\left(k_{law(1)}^{(i)}, ..., k_{law(4000)}^{(i)}\right)$. From each value of knowledge, the corresponding factual $\left(g_{law(j)}^{(i)}, l_{law(j)}^{(i)}\right)$ and counterfactual features $\left(\check{g}_{law(j)}^{(i)}, \check{l}_{law(j)}^{(i)}\right)$ are generated. The counterfactual fair representation is defined as $h_{CF}^{(i)} = \frac{1}{4000} \sum_{j=1}^{4000} \left[k_{law(j)}^{(i)}, r_{law}^{(i)}, \frac{g_{law(j)}^{(i)}+\check{g}_{law(j)}^{(i)}}{2}, \frac{l_{law(j)}^{(i)}+\check{l}_{law(j)}^{(i)}}{2}\right]$.

Table 3 displays the results of the CFR method and baselines in terms of the three metrics. The trend is very similar to our synthetic experiment. Again, the CF representation is the only method which can achieve perfect counterfactual fairness (with TE = 0). For the statistical parity, the CF representation improves A-Accuracy from 100% to 57.6% compared to the UF method. It is worth mentioning that the CF representation is much easier to train and generate without the complex and unstable adversarial process. In the experiment, the counterfactually fair representation only increases the MSE by 5% compared to the UF method while achieving two notions of fairness.

## 5 Conclusion

In this paper, we build a connection between counterfactual fairness, statistical parity, and individual fairness. In particular, we prove that under the Gaussian causal model, counterfactually fair representation satisfies statistical parity and individual fairness at the same time. We also prove that for a broader family of causal models, the counterfactually fair representation is independent of the sensitive attribute. On the other hand, we show that a predictor satisfying statistical parity or individual fairness generally may not satisfy counterfactual fairness. Several experiments on both synthetic and real-world datasets confirm our theoretical results. In particular, under conditions that we have in our theorems, we observe that a predictor trained on counterfactually fair representations can achieve statistical parity and individual fairness with a similar MSE level as the baselines.

**Limitation and Social Impact**

This work establishes the connection between CF achieved through CFR, SP, and IF, showing that counterfactually fair representation can be used to satisfy various types of fairness if the necessary assumptions are met. It is important to note that the discussion in this paper focuses on counterfactual fairness achieved through counterfactually fair representation, rather than abstract counterfactual fairness. Since the accurate construction of such representations depends on known SCMs, our conclusions may not be valid with an incorrect SCM. Additionally, the assumptions outlined in Section 3 require verification and might not hold in every scenario.

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

# A  Proofs

## A.1  Theorem 1

*Proof.* Suppose $U$ is $d_u$ dimensional and $X$ is $d_x$ dimensional, we know the probability density functions are

$$p_U(u) = \frac{1}{(2\pi)^{\frac{d_u}{2}} |\Sigma_u|^{\frac{1}{2}}} e^{-\frac{1}{2}(u-\mu_u)^{\mathrm{T}}\Sigma_u^{-1}(u-\mu_u)}, \tag{15}$$

$$p_{X|U,A}(x|u,a) = \frac{1}{(2\pi)^{\frac{d_x}{2}}|\Sigma_x|^{\frac{1}{2}}} e^{-\frac{1}{2}(x-W_uu-f_a(a)-b)^{\mathrm{T}}\Sigma_x^{-1}(x-W_uu-f_a(a)-b)}. \tag{16}$$

Based on Bayes theorem, we have the posterior distribution

$$p_{U|X,A}(u|x,a) = \frac{p_{U,A,X}(u,a,x)}{p_{A,X}(a,x)} = \frac{p_{X|A,U}(x|a,u)p_{A,U}(a,u)}{\int_U p_{X|A,U}(x|a,u)p_{A,U}(a,u)\mathrm{d}u}. \tag{17}$$

If $A$ and $U$ are independent,

$$
\begin{aligned}
p_{U|X,A}(u|x,a) &= \frac{e^{-\frac{1}{2}[(u-\mu_u)^{\mathrm{T}}\Sigma_u^{-1}(u-\mu_u)+(x-W_uu-f_a(a)-b)^{\mathrm{T}}\Sigma_x^{-1}(x-W_uu-f_a(a)-b)]}}{\int_{-\infty}^{\infty} e^{-\frac{1}{2}[(u-\mu_u)^{\mathrm{T}}\Sigma_u^{-1}(u-\mu_u)+(x-W_uu-f_a(a)-b)^{\mathrm{T}}\Sigma_x^{-1}(x-W_uu-f_a(a)-b)]}\mathrm{d}u} \\
&= \frac{e^{-\frac{1}{2}(u-\mu)^{\mathrm{T}}\Sigma^{-1}(u-\mu)}}{\int_{-\infty}^{\infty} e^{-\frac{1}{2}(u-\mu)^{\mathrm{T}}\Sigma^{-1}(u-\mu)}\mathrm{d}u} \\
&= \frac{1}{(2\pi)^{\frac{d_u}{2}}|\Sigma|^{-1}} e^{-\frac{1}{2}(u-\mu)^{\mathrm{T}}\Sigma^{-1}(u-\mu)},
\end{aligned}
\tag{18}
$$

where

$$\mu = (W_u^{\mathrm{T}}\Sigma_x^{-1}W_u + \Sigma_u^{-1})^{-1}[W_u^{\mathrm{T}}\Sigma_x^{-1}(x-f_a(a)-b) + \Sigma_u^{-1}\mu_u], \tag{19}$$

$$\Sigma^{-1} = W_u^T\Sigma_x^{-1}W_u + \Sigma_u^{-1}. \tag{20}$$

**Case 1**: When we use $H(x,a)$ as the representation for $(x,a)$, $H$ is a random variable. And $H(x,a)$ is a function of $U$. With the definition introduced in Dwork et al. (2012), we know that the total variation between the distribution associated with $H(x,a)$ and $H(x',a')$ can be used to describe the distance between the representations. And we have

$$d(H(x,a),H(x',a')) \le D_{tv}(p_{U|X,A}(x,a),p_{U|X,A}(x',a')). \tag{21}$$

$D_{tv}$ is the total variation distance between the two distributions[13]. By Pinsker's inequality:

$$D_{tv}(p_{U|X,A}(x,a),p_{U|X,A}(x',a')) \le \sqrt{\frac{1}{2}D_{KL}(p_{U|X,A}(x,a)||p_{U|X,A}(x',a'))}. \tag{22}$$

$D_{KL}$ is the Kullback-Leibler divergence. Because the divergence of the two Gaussian distribution is

$$
\begin{aligned}
D_{KL}(p_{U|X,A}(x,a)||p_{U|X,A}(x',a')) &= \frac{1}{2}\left[\ln\frac{\Sigma}{\Sigma} - n_x + \mathrm{tr}[\Sigma^{-1}\Sigma] + (\mu'-\mu)^{\mathrm{T}}\Sigma^{-1}(\mu'-\mu)\right] \\
&= \frac{1}{2}\left[(\mu'-\mu)^{\mathrm{T}}\Sigma^{-1}(\mu'-\mu)\right],
\end{aligned}
\tag{23}
$$

where

$$\mu' = (W_u^{\mathrm{T}}\Sigma_x^{-1}W_u + \Sigma_u^{-1})^{-1}[W_u^{\mathrm{T}}\Sigma_x^{-1}(x'-f_a(a')-b) + \Sigma_u^{-1}\mu_u]. \tag{24}$$

Let

$$C = (W_u^{\mathrm{T}}\Sigma_x^{-1}W_u + \Sigma_u^{-1})^{-1}W_u^{\mathrm{T}}\Sigma_x^{-1}, \tag{25}$$

we can get

$$
\begin{aligned}
&D_{KL}(p_{U|X,A}(x,a)||p_{U|X,A}(x',a')) \\
=&\frac{1}{2}\left[[C((x'-x)-(f_a(a')-f_a(a)))]^{\mathrm{T}}\Sigma^{-1}[C((x'-x)-(f_a(a')-f_a(a)))]\right] \\
=&\frac{1}{2}\left[[(x'-x)+(f_a(a')-f_a(a))]^{\mathrm{T}}(C^{\mathrm{T}}\Sigma^{-1}C)[(x'-x)+(f_a(a')-f_a(a))]\right].
\end{aligned}
\tag{26}
$$

---

[13]Here $H(x,a)$ is a random variable which is a function of $U$. We utilize the property that the total variation after a functional transformation will always decrease. The property is proved by Lemma 1.

Since $C^{\mathrm{T}}\Sigma^{-1}C$ is symmetric, we have

$$D_{KL}(p_{U|X,A}(x,a)||p_{U|X,A}(x',a')) \leq \frac{1}{2}\left\|C^{\mathrm{T}}\Sigma^{-1}C\right\|_2^2\left\|(x'-x)-(f_a(a')-f_a(a))\right\|_2^2$$

$$\leq \left\|C^{\mathrm{T}}\Sigma^{-1}C\right\|_2^2[\|x'-x\|_2^2+\|f_a(a')-f_a(a)\|_2^2]. \tag{27}$$

Because $f_a$ is Lipschitz continuous,

$$\|f_a(a')-f_a(a)\|_2^2 \leq L_a\|a'-a\|_2^2, \tag{28}$$

so,

$$D_{KL}(p_{U|X,A}(x,a)||p_{U|X,A}(x',a')) \leq \left\|C^{\mathrm{T}}\Sigma^{-1}C\right\|_2^2[\|x'-x\|_2^2+L_a\|a'-a\|_2^2]$$

$$\leq \left\|C^{\mathrm{T}}\Sigma^{-1}C\right\|_2^2 \cdot \max\{1,L_a\}\left\|(x,a)-(x',a')\right\|_2^2, \tag{29}$$

which is to say

$$d(H(x,a),H(x',a')) \leq L_1\left\|(x,a)-(x',a')\right\|_2, \tag{30}$$

where

$$L_1 = \sqrt{\frac{1}{2}\left\|C^{\mathrm{T}}\Sigma^{-1}C\right\|_2^2 \cdot \max\{1,L_a\}}. \tag{31}$$

**Case 2**: When we use $r(x,a)$ as the counterfactually fair representation, given x, a, we have

$$r(x,a) = \int_{-\infty}^{\infty}\frac{1}{(2\pi)^{\frac{n_u}{2}}|\Sigma|^{-1}}e^{-\frac{1}{2}(u-\mu)^{\mathrm{T}}\Sigma^{-1}(u-\mu)}\left[s\left(\mathbb{E}\left[\check{x}[a^{[1]}]|u\right],...,\mathbb{E}\left[\check{x}[a^{[|\mathcal{A}|]}]|u\right]\right),u\right]\mathrm{d}u. \tag{32}$$

Because we know that $\mathbb{E}\left[\check{x}[a^{[i]}]|u\right]$ is a function of $u$, we write $s(\mathbb{E}\left[\check{x}[a^{[1]}]|u\right],...,\mathbb{E}\left[\check{x}[a^{[|\mathcal{A}|]}]|u\right])$ as $\bar{s}(u)$. Then the representation can be divided into two parts:

$$r_x(x,a) = \int_{-\infty}^{\infty}\frac{1}{(2\pi)^{\frac{n_u}{2}}|\Sigma|^{-1}}e^{-\frac{1}{2}(u-\mu)^{\mathrm{T}}\Sigma^{-1}(u-\mu)}\bar{s}(u)\mathrm{d}u, \tag{33}$$

$$r_u(x,a) = \int_{-\infty}^{\infty}\frac{1}{(2\pi)^{\frac{n_u}{2}}|\Sigma|^{-1}}e^{-\frac{1}{2}(u-\mu)^{\mathrm{T}}\Sigma^{-1}(u-\mu)}u\mathrm{d}u. \tag{34}$$

For the part $u$, we have

$$\int_{-\infty}^{\infty}\frac{1}{(2\pi)^{\frac{n_u}{2}}|\Sigma|^{-1}}e^{-\frac{1}{2}(u-\mu)^{\mathrm{T}}\Sigma^{-1}(u-\mu)}u\mathrm{d}u = \mu. \tag{35}$$

Suppose $\bar{s}(u)$ is Lipschitz continuous w.r.t. $u$, which is to say

$$\left\|\bar{s}(u)-\bar{s}(u')\right\|_2 \leq L_u\left\|u-u'\right\|_2, \tag{36}$$

we have

$$\left\|r_x(x,a)-r_x(x',a')\right\|_2$$

$$=\left\|\int_{\infty}^{\infty}\frac{1}{(2\pi)^{\frac{n_u}{2}}|\Sigma|^{-1}}e^{-\frac{1}{2}(u-\mu)^{\mathrm{T}}\Sigma^{-1}(u-\mu)}[\bar{s}(u)-\bar{s}(u+\mu'-\mu)]\right\|_2\mathrm{d}u$$

$$=\int_{-\infty}^{\infty}\left\|\bar{s}(u)-\bar{s}(u+\mu'-\mu)\right\|_2\frac{1}{(2\pi)^{\frac{n_u}{2}}|\Sigma|^{-1}}e^{-\frac{1}{2}(u-\mu)^{\mathrm{T}}\Sigma^{-1}(u-\mu)}\mathrm{d}u$$

$$\leq L_u\int_{-\infty}^{\infty}\left\|\mu'-\mu\right\|_2\frac{1}{(2\pi)^{\frac{n_u}{2}}|\Sigma|^{-1}}e^{-\frac{1}{2}(u-\mu)^{\mathrm{T}}\Sigma^{-1}(u-\mu)}\mathrm{d}u$$

$$=L_u\left\|\mu'-\mu\right\|_2. \tag{37}$$

Therefore,

$$\|r(x,a) - r(x',a')\|_2 \leq \|r_u(x,a) - r_u(x',a')\|_2 + \|r_x(x,a) - r_x(x',a')\|_2$$
$$\leq (1 + L_u)\|\mu' - \mu\|_2. \tag{38}$$

From case 1, we know,

$$\|\mu' - \mu\|_2 \leq \sqrt{2\|C^{\mathrm{T}}C\|_2^2 \cdot \max\{1, L_a\}}\,\|(x,a) - (x',a')\|_2, \tag{39}$$

so we can get

$$\|r(x,a) - r(x',a')\|_2 \leq L_2\,\|(x,a) - (x',a')\|_2, \tag{40}$$

where

$$L_2 = (1 + L_u)\sqrt{\|C^{\mathrm{T}}C\|_2^2 \cdot \max\{1, L_a\}}. \tag{41}$$

$$\square$$

## A.2 Theorem 3

*Proof.* **Case 1**: When we use $H(X, A)$ as the representation,

$$H = [\bar{s}(U), U]. \tag{42}$$

Given $x_s$, suppose $\bar{s}(u) = x_s$ has $k$ solutions in $u$, say $u_1, u_2, ..., u_k$. The probability density function of $X_s$ is given by

$$p_{\bar{s}(U)}(x_s) = \sum_{i=1}^{k} p_U(u_i)\left|\frac{\mathrm{d}\bar{s}(u_i)}{\mathrm{d}u}\right|^{-1}. \tag{43}$$

By the independence of $U$ and $A$,

$$p_{U,A}(u,a) = p_U(u)p_A(a). \tag{44}$$

This holds for all $u$ and $a$. So the joint probability density function of $\bar{s}(U)$ and $A$ can be expressed as:

$$p_{\bar{s}(U),A}(x_s,a) = \sum_{i=1}^{k} p_U(u_i)\left|\frac{\mathrm{d}\bar{s}(u_i)}{\mathrm{d}u}\right|^{-1} f_A(a). \tag{45}$$

So we have

$$p_{\bar{s}(U),A}(x_s,a) = p_{\bar{s}(U)}(x_s)p_A(a), \tag{46}$$

which is to say $\bar{s}(U) \perp A$.

Because

$$p_{H,A}(h,a) = p_{\bar{s}(U),U,A}(x_s,u,a) = p_{\bar{s}(U),U}(x_s,u)p_A(a), \tag{47}$$

we have $H(X, A) \perp A$.

**Case 2**: When we use r(X, A) as representation, we denote,

$$r(X,A) = [r_x(X,A), r_u(X,A)]. \tag{48}$$

The joint probability density function is

$$p_{r,A}(r,a) = p_{r_x,r_u,A}(r_x,r_u,a). \tag{49}$$

Suppose $r_x(x,a) = r_{x0}, r_u(x,a) = r_{u0}$ has $k$ solutions, say $(x_1, a_1), ..., (x_k, a_k)$. The probability density function of $(r, A)$ is given by

$$p_{r_x, r_u, A}(r_{x0}, r_{u0}, a) = \sum_{i=1}^{k} p_{X,A}(x_i, a_i) \cdot \begin{vmatrix} \frac{\partial r_x(x_i, a_i)}{\partial x} & \frac{\partial r_x(x_i, a_i)}{\partial a} \\ \frac{\partial r_u(x_i, a_i)}{\partial x} & \frac{\partial r_u(x_i, a_i)}{\partial a} \end{vmatrix}. \tag{50}$$

For any $a_1, a_2$, we have

$$p_{r_x, r_u, A}(r_{x0}, r_{u0}, a_1) = \sum_{i=1}^{k} \mathbf{1}(a_i = a_1) \cdot p_{X,A}(x_i, a_i) \cdot \begin{vmatrix} \frac{\partial r_x(x_i, a_i)}{\partial x} & \frac{\partial r_x(x_i, a_i)}{\partial a} \\ \frac{\partial r_u(x_i, a_i)}{\partial x} & \frac{\partial r_u(x_i, a_i)}{\partial a} \end{vmatrix}, \tag{51}$$

$$p_{r_x, r_u, A}(r_{x0}, r_{u0}, a_2) = \sum_{i=1}^{k} \mathbf{1}(a_i = a_2) \cdot p_{X,A}(x_i, a_i) \cdot \begin{vmatrix} \frac{\partial r_x(x_i, a_i)}{\partial x} & \frac{\partial r_x(x_i, a_i)}{\partial a} \\ \frac{\partial r_u(x_i, a_i)}{\partial x} & \frac{\partial r_u(x_i, a_i)}{\partial a} \end{vmatrix}. \tag{52}$$

Now for any $x_i$ which makes $a_i = a_1$, we have

$$x_i - f_a(a_1) = (W_u^{\mathrm{T}} \Sigma_x^{-1})[(W_u^{\mathrm{T}} \Sigma_x^{-1} W_u + \Sigma_u^{-1}) - \Sigma_u^{-1} \mu_u] + b, \tag{53}$$

we can $x_j$ which makes $a_j = a_2$ satisfies

$$x_j - f_a(a_2) = (W_u^{\mathrm{T}} \Sigma_x^{-1})[(W_u^{\mathrm{T}} \Sigma_x^{-1} W_u + \Sigma_u^{-1}) - \Sigma_u^{-1} \mu_u] + b. \tag{54}$$

Because

$$p_{X,A}(x,a) = \int_{-\infty}^{\infty} e^{-\frac{1}{2}(u-\mu)^{\mathrm{T}} \Sigma^{-1}(u-\mu)} \mathrm{d}u \; p_A(a), \tag{55}$$

where

$$\mu = (W_u^{\mathrm{T}} \Sigma_x^{-1} W_u + \Sigma_u^{-1})^{-1}[W_u^{\mathrm{T}} \Sigma_x^{-1}(x - f_a(a) - b) + \Sigma_u^{-1} \mu_u], \tag{56}$$

we have

$$p_{X,A}(x_i, a_1) p_A(a_1) = p_{X,A}(x_i, a_2) p_A(a_2). \tag{57}$$

According to the definition of $r_x(x,a)$ and $r_u(x,a)$, we know that

$$\frac{\partial r_x(x,a)}{\partial x} = \int_{-\infty}^{\infty} e^{-\frac{1}{2}(u-\mu)^{\mathrm{T}} \Sigma^{-1}(u-\mu)} (u-\mu)^{\mathrm{T}} \Sigma^{-1} \frac{\partial \mu}{\partial x} \bar{s}(u) \mathrm{d}u, \tag{58}$$

where

$$\frac{\partial \mu}{\partial x} = (W_u^{\mathrm{T}} \Sigma_x^{-1} W_u + \Sigma_u^{-1})^{-1} W_u^{\mathrm{T}} \Sigma_x^{-1}, \tag{59}$$

and

$$\frac{\partial r_x(x,a)}{\partial a} = \int_{-\infty}^{\infty} e^{-\frac{1}{2}(u-\mu)^{\mathrm{T}} \Sigma^{-1}(u-\mu)} (u-\mu)^{\mathrm{T}} \Sigma^{-1} \frac{\partial \mu}{\partial a} \bar{s}(u) \mathrm{d}u, \tag{60}$$

where

$$\frac{\partial \mu}{\partial a} = (W_u^{\mathrm{T}} \Sigma_x^{-1} W_u + \Sigma_u^{-1})^{-1} W_u^{\mathrm{T}} \Sigma_x^{-1} \frac{\mathrm{d}f_a(a)}{\mathrm{d}a}, \tag{61}$$

and

$$\frac{\partial r_u(x,a)}{\partial x} = \frac{\partial \mu}{\partial x} = (W_u^{\mathrm{T}}\Sigma_x^{-1}W_u + \Sigma_u^{-1})^{-1}W_u^{\mathrm{T}}\Sigma_x^{-1}, \tag{62}$$

and

$$\frac{\partial r_u(x,a)}{\partial a} = \frac{\partial \mu}{\partial x} = (W_u^{\mathrm{T}}\Sigma_x^{-1}W_u + \Sigma_u^{-1})^{-1}W_u^{\mathrm{T}}\Sigma_x^{-1}\frac{\mathrm{d}f_a(a)}{\mathrm{d}a}. \tag{63}$$

When $\frac{\mathrm{d}f_a(a)}{\mathrm{d}a}$ is a constant, we have

$$\frac{\partial r_x(x_i,a_1)}{\partial x} = \frac{\partial r_x(x_j,a_2)}{\partial x}, \quad \frac{\partial r_x(x_i,a_1)}{\partial a} = \frac{\partial r_x(x_j,a_2)}{\partial a},$$
$$\frac{\partial r_u(x_i,a_1)}{\partial x} = \frac{\partial r_u(x_j,a_2)}{\partial x}, \quad \frac{\partial r_u(x_i,a_1)}{\partial a} = \frac{\partial r_u(x_j,a_2)}{\partial a}, \tag{64}$$

which is to say,

$$p_{X,A}(x_i,a_1)p_A(a_1) \cdot \begin{vmatrix} \frac{\partial r_x(x_i,a_1)}{\partial x} & \frac{\partial r_x(x_i,a_1)}{\partial a} \\ \frac{\partial r_u(x_i,a_1)}{\partial x} & \frac{\partial r_u(x_i,a_1)}{\partial a} \end{vmatrix} = p_{X,A}(x_j,a_2)p_A(a_2) \cdot \begin{vmatrix} \frac{\partial r_x(x_j,a_2)}{\partial x} & \frac{\partial r_x(x_j,a_2)}{\partial a} \\ \frac{\partial r_u(x_j,a_2)}{\partial x} & \frac{\partial r_u(x_j,a_2)}{\partial a} \end{vmatrix}. \tag{65}$$

So, for any $a_1, a_2$, we have

$$p_{r_x,r_u,A}(r_{x0}, r_{u0}, a_1)p_A(a_1) = p_{r_x,r_u,A}(r_{x0}, r_{u0}, a_2)p_A(a_2). \tag{66}$$

Because $P_A(a)$ is a uniform distribution, we have that

$$p_{r_x,r_u|A}(r_{x0}, r_{u0}|a_1) = \frac{p_{r_x,r_u,A}(r_{x0}, r_{u0}, a_1)}{p_A(a_1)} = \frac{p_{r_x,r_u|A}(r_{x0}, r_{u0}|a_1)p_A(a_1)}{p_A^2(a_1)}$$
$$= \frac{p_{r_x,r_u|A}(r_{x0}, r_{u0}|a_2)p_A(a_2)}{p_A^2(a_2)} = \frac{p_{r_x,r_u,A}(r_{x0}, r_{u0}, a_2)}{p_A(a_2)} = p_{r_x,r_u|A}(r_{x0}, r_{u0}|a_2), \tag{67}$$

which is to say $p_{r_x,r_u|A}(r_{x0}, r_{u0}|a)$ is irrelevant to $a$. So, we have $r(X,A) \perp A$. $\qquad\square$

## A.3 Corollary 2

*Proof.* When we use $H$ as the input of $g$, because of the total variation has the property (seen in Lemma 1) that

$$d(g(H(x,a), H(x',a')) \le d(H(x,a), H(x',a')), \tag{68}$$

we have

$$d(g(H(x,a), H(x',a')) \le L_1 \|(x,a) - (x',a')\|_2 \tag{69}$$

When we use $r(x,a)$ as the input of $g$, similarly we have

$$\|g(r(x,a)) - g(r(x',a'))\|_2 \le L_g \|r(x,a) - r(x',a')\|_2 \le L_g L_2 \|(x,a) - (x',a')\|_2 \tag{70}$$

$$\square$$

## A.4 Theorem 4

*Proof.* When the SCM is determinstic, we know that $X$ are determined by $U$ and $A$. We denote is as $X = f(U,A)$. Therefore, we can write $s\left(\mathbb{E}\left[\check{X}[\check{a}^{[1]}]|U\right], ..., \mathbb{E}\left[\check{X}[\check{a}^{|\mathcal{A}|}]|\right]|U\right)$ as $s\left(f(U,a^{[1]}), ..., f(U,a^{|\mathcal{A}|})\right)$.

**Case 1**: When we use $H(X, A)$ as representation,

$$H = [s\left(f(U, a^{[1]}), ..., f(U, a^{|\mathcal{A}|})\right), U] \tag{71}$$

We still denote $s\left(f(U, a^{[1]}), ..., f(U, a^{|\mathcal{A}|})\right)$ as $\bar{s}(U)$. Because $U$ is independent of $A$, we have $\bar{s}(U)$ is also independent of $U$ (seen in Appendix A.2). Therefore, we have $H \perp A$.

**Case 2**: When we use $r(X, A)$ as representation, suppose we have the causal model consisting of $U$, $A$, $X$ with domain $\mathbb{U}$, $\mathbb{A}$ and $\mathbb{X}$. The prior distribution of $U$ and $A$ are encoded in the probability density functions $p(U)$ and $p(A)$.

Because $U \perp A$, the joint distribution of $U, A, X$ can be written as

$$p_{U,A,X}(u, a, x) = p_U(u)p_A(a)\delta(x - f(u, a)). \tag{72}$$

From Bayes theorem, we know

$$p_{U|A,X}(u|x, a) = \frac{p_{U,A,X}(u, a, x)}{p_{A,X}(x, a)} = \frac{p_U(u)p_A(a)\delta(x - f(u, a))}{\int_{\mathbb{U}} p_U(u)p_A(a)\delta(x - f(u, a))\mathrm{d}u}. \tag{73}$$

With the definition of $r(X, A)$, we can know

$$p_{r|A,X}(r_0|a, x) = \delta\left[\int_{\mathbb{U}} \frac{p_U(u)p_A(a)\delta(x - f(u, a)}{\int_{\mathbb{U}} p_U(u)p_A(a)\delta(x - f(u, a))\mathrm{d}u}[\bar{s}(u), u]\mathrm{d}u - r_0\right]. \tag{74}$$

The conditional distribution of $r(X, A)$ on $A$ is

$$p_{r|A}(r_0|a) = \int_{\mathbb{X}} p_{r|X,A}(r_0|x, a)p_{X|A}(x|a)\mathrm{d}x. \tag{75}$$

And

$$p_{X|A}(x|a) = \frac{p_{X,A}(x, a)}{p_A(a)} = \frac{\int_{\mathbb{U}} p_A(a)p_U(u)\delta(x - f(u, a))\mathrm{d}u}{p_A(a)}. \tag{76}$$

As a result,

$$
\begin{aligned}
p_{r|A}(r_0|a) &= \int_{\mathbb{X}} p_{r|A,X}(r_0|a, x)p_{X|A}(x|a)\mathrm{d}x \\
&= \int_{\mathbb{X}} \delta\left[\int_{\mathbb{U}} \frac{p_U(u)p_A(a)\delta(x - f(u, a))}{\int_{\mathbb{U}} p_U(u)p_A(a)\delta(x - f(u, a))\mathrm{d}u}[\bar{s}(u), u]\mathrm{d}u - r_0\right] \frac{\int_{\mathbb{U}} p_A(a)p_U(u)\delta(x - f(u, a))\mathrm{d}u}{p_A(a)}\mathrm{d}x.
\end{aligned} \tag{77}
$$

Because $U$ follows a uniform distribution, $p_{r|A}(r_0|a)$ can be simplified as

$$p_{r|A}(r_0|a) = \int_{\mathbb{X}} \delta\left[\int_{\mathbb{U}} \frac{\delta(x - f(u, a))}{\int_{\mathbb{U}} \delta(x - f(u, a))\mathrm{d}u}[\bar{s}(u), u]\mathrm{d}u - r_0\right] \int_{\mathbb{U}} p_U(u)\delta(x - f(u, a))\mathrm{d}u\mathrm{d}x. \tag{78}$$

Suppose given $x, a$, the solution of $x = f(u, a)$ is $\mathbb{U}_a^x$. Denote the size of $\mathbb{U}_a^x$ as $\Omega_a^x$, since $p_U(u)$ is a uniform distribution, we have

$$\int_{\mathbb{U}_a^x} p_U(u)\delta(x - f(u, a))\mathrm{d}u = \Omega_a^x \tag{79}$$

Given $r_0$, we assume that $\mathcal{X} = \{x_1, x_2, ..., x_k\}$ is the set of solutions making that

$$\int_{\mathbb{U}} \frac{\delta(x - f(u, a))}{\int_{\mathbb{U}} \delta(x - f(u, a))\mathrm{d}u}[\bar{s}(u), u]\mathrm{d}u = r_0, \tag{80}$$

so, we have

$$p_{r|A}(r_0|a) = \sum_{x \in \mathcal{X}} \Omega_a^x \tag{81}$$

We consider $p_{r|A}(r_0|a')$. Because of the condition 3 in Theorem 4, we know that

$$f(u_1, a) = f(u_2, a) \Leftrightarrow f(u_1, a') = f(u_2, a'), \tag{82}$$

so, we can divide the space $\mathbb{U}$ based on $\mathbb{U}_a^x$ to make that

$$f(u, a') = x' \quad \forall u \in \mathbb{U}_a^x. \tag{83}$$

Therefore, for $x \in \mathcal{X}$, if Eq. 80 holds, there exists $x'$ satisfies

$$f(u, a') = x' \quad \forall u \in \mathbb{U}_a^x, \tag{84}$$

and

$$\int_{\mathbb{U}} \frac{\delta(x' - f(u, a'))}{\int_{\mathbb{U}} \delta(x' - f(u, a')) \mathrm{d}u} [\bar{s}(u), u] \mathrm{d}u = r_0. \tag{85}$$

Therefore, we have

$$p_{r|A}(r_0|a') = \sum_{x \in \mathcal{X}} \Omega_a^x. \tag{86}$$

So, we prove that $p_{r|A}(r_0|a) = p_{r|A}(r_0|a')$ for any $a, a' \in \mathbb{A}$, which is to say $r(X, A) \perp A$. $\qquad \square$

## B  Lemma for Proof

**Lemma 1.** For any random variable $U$ and $U'$, the probability distributions of $U$ and $U'$ are $p_U(u)$ and $p_{U'}(u)$. Let $\mathcal{F}$ be an arbitrary function, $V = \mathcal{F}(U)$ and $V' = \mathcal{F}(U')$ are two random variables[14] with distributions $p_V(v)$ and $P_{V'}(v)$. Then the total variation satisfies

$$d(V, V') \leq d(U, U'). \tag{87}$$

*Proof.* For two probability distributions $p_U(u)$ and $p_{U'}(u)$, assume the domain space of $U$ is $\mathbb{U}$, the total variation between them is

$$D_{tv}(p_U, p_{U'}) = \frac{1}{2} \int_{u \in \mathbb{U}} |p_U(u) - p_{U'}(u)| \, \mathrm{d}u. \tag{88}$$

For any $v \in \mathbb{V}$, we use subspace $\mathbb{U}^v$ to denote the subspace that for any $u \in \mathbb{U}^v$, there is $\mathcal{F}(u) = v$. Then we have

$$p_V(v) = \int_{u \in \mathbb{U}^v} p_U(u) \mathrm{d}u, \tag{89}$$

and

$$p_V(v) = \int_{u \in \mathbb{U}^v} p_{U'}(u) \mathrm{d}u. \tag{90}$$

Consider the total variation distance for the induced distributions,

$$D_{tv}(p_V, p_{V'}) = \frac{1}{2} \int_{\mathbb{V}} \left| \int_{u \in \mathbb{U}^v} p_U(u) \mathrm{d}u - \int_{u \in \mathbb{U}^v} p_{U'}(u) \mathrm{d}u \right| \mathrm{d}v. \tag{91}$$

---

[14]It should be noticed that $U$ and $V$ in the Lemma and its proof are not mean they are unobservable and observable variables in a SCM. They are only used to represents arbitrary random variables in this section.

Using the triangle inequality for absolute values, we have

$$\left| \int_{u \in \mathbb{U}^v} p_U(u) \mathrm{d}u - \int_{u \in \mathbb{U}^v} p_{U'}(u) \mathrm{d}u \right| \leq \int_{u \in \mathbb{U}^v} |p_U(u) - p_{U'}(u)| \, \mathrm{d}u. \tag{92}$$

Therefore,

$$\int_{\mathbb{V}} \left| \int_{u \in \mathbb{U}^v} p_U(u) \mathrm{d}u - \int_{u \in \mathbb{U}^v} p_{U'}(u) \mathrm{d}u \right| \mathrm{d}v \leq \int_{\mathbb{V}} \int_{u \in \mathbb{U}^v} |p_U(u) - p_{U'}(u)| \, \mathrm{d}u \mathrm{d}v. \tag{93}$$

Notice that each $u$ appears in only one of the $\mathbb{U}^v$. Thus, we have,

$$\int_{\mathbb{V}} \int_{u \in \mathbb{U}^v} |p_U(u) - p_{U'}(u)| \, \mathrm{d}u \mathrm{d}v = \int_{u \in \mathbb{U}} |p_U(u) - p_{U'}(u)| \, \mathrm{d}u, \tag{94}$$

which means,

$$\int_{\mathbb{V}} \left| \int_{u \in \mathbb{U}^v} p_U(u) \mathrm{d}u - \int_{u \in \mathbb{U}^v} p_{U'}(u) \mathrm{d}u \right| \mathrm{d}v \leq \int_{u \in \mathbb{U}} |p_U(u) - p_{U'}(u)| \, \mathrm{d}u. \tag{95}$$

So we have $D_{tv}(p_V, p_{V'}) \leq D_{tv}(p_U, p_{U'})$. $\qquad \square$

## C   Discussion on Gamma distributions

**Theorem 5.** Given a structural causal model (SCM) $\mathcal{M}(U, V, F)$, where the following conditions holds:

1. $P(U)$ is a Gamma distribution

$$U \sim \mathrm{Gamma}(\alpha, \beta). \tag{96}$$

2. The structural function for $X$ is given by,

$$X \sim \mathrm{Exponential}(W_u u), \tag{97}$$

where $W_u > 0$.

3. $A \perp U$.

For the counterfactually fair representation $r(x, a) = \mathbb{E}_{U \sim \mathrm{Pr}_{\mathcal{M}}\{U | X = x, A = a\}}[U]$, we have

$$\|r(x, a) - r(x', a')\|_2 \leq L^{\mathrm{gamma}} \|(x, a) - (x', a')\|_2 \ \forall x, x', a, a'. \tag{98}$$

where $L^{\mathrm{gamma}} = \left\| \frac{(\alpha+1) W_u}{\beta^2} \right\|_2$.

*Proof.* Because $P(U)$ satisfies the Gamma distribution, we know that

$$p_U(u) = \frac{\beta^\alpha}{\Gamma(\alpha)} u^{\alpha-1} e^{-\beta u}, \ u > 0. \tag{99}$$

The conditional distribution of $P_{X|U,A}(x|u, a)$ is

$$p_{X|U,A}(x|u, a) = f_a(a) W_u u e^{-W_u u x}. \tag{100}$$

Since $A \perp U$, we know the posterior distribution is

$$p_{U|X,A}(u|x, a) = \frac{u^\alpha e^{-(\beta + W_u x) u}}{\int_0^\infty u^\alpha e^{-(\beta + W_u x) u} \mathrm{d}u} = \frac{(\beta + W_u x)^{\alpha+1}}{\Gamma(\alpha+1)} u^\alpha e^{-(\beta + W_u x) u}. \tag{101}$$

Therefore, the posterior distribution of $U$ is a Gamma distribution $\text{Gamma}(\alpha + 1, \beta + W_u x)$. The representation

$$r(x, a) = \mathbb{E}_{U \sim \text{Pr}_{\mathcal{M}}\{U|X=x, A=a\}}[U] = \frac{\alpha + 1}{\beta + W_u x}. \tag{102}$$

We have

$$\begin{aligned}
\|r(x, a) - r(x', a')\|_2 &= \left\| \frac{\alpha + 1}{\beta + W_u x} - \frac{\alpha + 1}{\beta + W_u x'} \right\|_2 \\
&= \|(\alpha + 1)W_u\|_2 \left\| \frac{x' - x}{(\beta + W_u x)(\beta + W_u x')} \right\|_2 \\
&\leq \left\| \frac{(\alpha + 1)W_u}{\beta^2} \right\|_2 \|x' - x\|_2. \tag{103}
\end{aligned}$$

Therefore, $L^{\text{gamma}} = \left\| \frac{(\alpha+1)W_u}{\beta^2} \right\|_2$. $\qquad\qquad\qquad\qquad\qquad\qquad\qquad\qquad\qquad\qquad\qquad\qquad\square$

## D  Counter Example $A \perp U$ does not imply SP

**Example 4.** Suppose an causal model consists of $U, X, A$. The prior distribution of $U$ is

$$\text{Pr}\{U = -1\} = 0.4, \quad \text{Pr}\{U = 0\} = 0.3, \quad \text{Pr}\{U = 1\} = 0.3.$$

The distribution of $A$ is

$$\text{Pr}\{A = -1\} = 0.8, \quad \text{Pr}\{A = 1\} = 0.2. \tag{104}$$

$X$ is determined by $U$ and $A$ in this way:

$$X = \begin{cases} 1, & \text{if } U + A \geq 1 \\ 0, & \text{otherwise} \end{cases}. \tag{105}$$

Then we can have the joint distribution of $U, A, X$ as the Table 4. So for the observed data, we only have

| U | A | X | Pr |
|---|---|---|------|
| -1 | -1 | 0 | 0.32 |
| 0 | -1 | 0 | 0.24 |
| 1 | -1 | 0 | 0.24 |
| -1 | 1 | 0 | 0.08 |
| 0 | 1 | 1 | 0.06 |
| 1 | 1 | 1 | 0.06 |

Table 4: Joint Distribution

$\{A = -1, X = 0\}$, $\{A = 1, X = 0\}$ and $\{A = 1, X = 1\}$. We use $r(x, a)$ as the counterfactually fair representation for data $(x, a)$. Now for data $A = -1, X = 0$, the posterior distribution of $U$ is

$$\text{Pr}\{U = -1\} = 0.4, \quad \text{Pr}\{U = 0\} = 0.3, \quad \text{Pr}\{U = 1\} = 0.3.$$

When $U = -1$, $\frac{x + \check{x}}{2} = \frac{0 + 0}{2} = 0$. When $U = 0$, $\frac{x + \check{x}}{2} = \frac{0 + 1}{2} = 0.5$. When $U = 1$, $\frac{x + \check{x}}{2} = \frac{0 + 1}{2} = 0.5$. So $r = 0.3$.

For $A = 1, X = 0$, we know that the posterior distribution of $U$ is $\text{Pr}\{U = -1\} = 1$. So $r = 0$. And when $A = 1, X = 1$, the posterior distribution of $U$ is $\text{Pr}\{U = 0\} = \text{Pr}\{U = 1\} = 0.5$. So $r = 0.5$.

That means when $A = -1$, the distribution of $r(X, A = -1)$ is

$$\Pr\{r = 0.3\} = 1. \tag{106}$$

When $A = 1$, the distribution of $r(X, A = 1)$ is

$$\Pr\{r = 0\} = 0.4 \quad \Pr\{r = 1\} = 0.6. \tag{107}$$

Therefore, $\Pr\{r(x, A)|A = 0\} \geq \Pr\{r(x, A)|A = 1\}$. Statistical parity is not hold.

## E    Explanation of Example 2

For the given Gaussian causal model, we know that

$$U \sim \text{Gaussian}(\mu_u, \Sigma_u^2), \tag{108}$$

$$X \sim \text{Gaussian}(W_u U + W_a A + b, \Sigma_x^2), \tag{109}$$

and

$$Y \sim \text{Gaussian}(W_x X + b_x, \Sigma_x^2). \tag{110}$$

Hence, we know that $\mathbb{E}[Y] = W_x W_u \mathbb{E}[U] + W_x W_a A + W_x b + b_x$. So we can use the predictor $\hat{Y}(X, A) = W_x W_u \mathbb{E}_{U \sim \Pr_{\mathcal{M}}\{U|X,A\}}[U] + W_x W_a A + W_x b + b_x$. Then we have

$$||\hat{Y}(x, a) - \hat{Y}(x', a')||_2 = ||W_x W_u(\mathbb{E}_{U \sim \Pr_{\mathcal{M}}(U|X=x,A=a)}[U] - \mathbb{E}_{U \sim \Pr_{\mathcal{M}}(U|X=x',A=a')}[U]) + W_x W_a(a - a')||_2. \tag{111}$$

From Eq. 39, we know that

$$||\mathbb{E}_{U \sim \Pr_{\mathcal{M}}(U|X=x,A=a)}[U] - \mathbb{E}_{U \sim \Pr_{\mathcal{M}}(U|X=x',A=a')}[U]||_2 \leq \sqrt{2|C^T C|} \cdot \max\{1, W_a\} ||(x, a) - (x', a')||_2. \tag{112}$$

Therefore,

$$||\hat{Y}(x, a) - \hat{Y}(x', a')||_2 \leq \sqrt{2||C^T C||_2^2} ||W_x W_u||_2 \max\{1, ||W_a||_2\} ||(x, a) - (x', a')||_2 + ||W_x W_a||_2 ||a - a'||_2. \tag{113}$$

Because $C$ is a scalar in this example, $\sqrt{||C^T C||_2^2} = ||C||_2$. So,

$$||\hat{Y}(x, a) - \hat{Y}(x', a')||_2 \leq \sqrt{2}||W_x W_u C||_2 \max\{1, ||W_a||_2\} ||(x, a) - (x', a')||_2 + ||W_x W_u C|| ||C^{-1} W_u^{-1} W_x^{-1} W_x W_a||_2 ||a - a'||_2 \tag{114}$$

$$\leq 2||W_x W_u C||_2 (\max\{1, W_a\} + C^{-1} W_u^{-1} W_a) ||(x, a) - (x', a')||_2. \tag{115}$$

Because the prediction is fixed given $X = x, A = a$, we know that

$$\Pr\{\hat{Y}_{A \leftarrow a}(U) = \hat{y}|X = x, A = a\} = \delta\left(\hat{y} - W_x W_u \mathbb{E}_{U \sim \Pr_{\mathcal{M}}\{U|X=x,A=a\}}[U] - W_x W_a a - W_x b - b_x\right). \tag{116}$$

Since in the counterfactual world, we still have the same conditional distribution over $U$, we know that

$$\check{\hat{Y}}(x, a) = W_x W_u \mathbb{E}[U] + W_x W_a \check{a} + W_x b + b_x, \tag{117}$$

which is to say

$$\Pr\{\hat{Y}_{A \leftarrow \check{a}}(U) = \hat{y}|X = x, A = a\} = \delta\left(\hat{y} - W_x W_u \mathbb{E}_{U \sim \Pr_{\mathcal{M}}\{U|X=x,A=a\}}[U] - W_x W_a \check{a} - W_x b - b_x\right). \tag{118}$$

Counterfactual fairness is violated.

## F  Parameters for GCM Simulation

The parameter $W_u$ is

$$W_u = \begin{bmatrix} 0.88292245 & 1.29287793 & -0.82082917 & -0.70183216 & -0.39127569 \\ -0.60877832 & 1.13381659 & -1.49961377 & 0.54270513 & 1.38670018 \\ -0.57873781 & 1.47206281 & 1.15733417 & -0.34923801 & 0.81879373 \\ -0.82661724 & -1.10173591 & -0.46378857 & 1.35030991 & -0.45830616 \\ 0.04167563 & -1.00437605 & 0.86665223 & 0.83145994 & -0.70429947 \end{bmatrix} \quad (119)$$

The bias vector $b_u$ is

$$b = \begin{bmatrix} 0.34952773 & -0.51095599 & -1.25532379 & 0.73900495 & -0.8848992 \end{bmatrix}^{\mathrm{T}} \quad (120)$$

$f_a$ is simulated as $W_a a$, where $W_a$ is

$$W_a = \begin{bmatrix} -2.31705195 & -0.36172777 & 0.44253204 & -0.01319519 & 0.08048071 \end{bmatrix}^{\mathrm{T}} \quad (121)$$

The covariance matrix $\Sigma_x$ is set as

$$\Sigma_x = \begin{bmatrix} 6.08443145 & 3.04621728 \times 10^{-3} & -1.54271138 & -2.51012096 & 2.45237759 \\ 3.04621728 \times 10^{-3} & 9.70835803 & 2.44983711 & -4.81523612 \times 10^{-1} & 1.07998448 \\ -1.54271138 & 2.44983711 & 2.25988521 & -8.39006253 \times 10^{-1} & -2.28247159 \\ -2.51012096 & -4.81523612 \times 10^{-1} & -8.39006253 \times 10^{-1} & 2.72134995 & 1.15217315 \\ 2.45237759 & 1.07998448 & -2.28247159 & 1.15217315 & 8.79705458 \end{bmatrix} \quad (122)$$

When generating the target variable $Y$, we used the linear model $Y = W_x^{\mathrm{T}} X + b_x$ with

$$W_x = \begin{bmatrix} -1.22783934 & 0.68714368 & 0.52803583 & -0.96272343 & 0.62690475 \end{bmatrix}^{\mathrm{T}} \quad (123)$$

$$b_x = -0.13026780 \quad (124)$$

## G  Synthetic Parameters

The parameters for generating $X_1$ are

$$b_1 = 0.1 \quad w_1^A = [0.2, 0.1] \quad w_1^X = [0.3, 0.4, 0.7, 0.1, 0.2, 0.4, 0.5, 0.2] \quad w_1^U = 0.3 \quad (125)$$

$X_2$ related parameters are

$$b_2 = 0.3 \quad w_1^A = [0.4, 0.3] \quad w_1^X = [0.1, 0.5, 0.6, 0.4, 0.3, 0.7, 0.8, 0.6] \quad w_1^U = 0.6 \quad (126)$$

To generate the target attribute $Y$,

$$w_Y^A = [0.5, 0.2] \quad w_1^X = [0.6, 0.7, 0.2, 0.3, 0.1, 0.6, 0.8, 0.4] \quad w_1^U = 0.5 \quad (127)$$

## H  More Implementation Details

The parameters and structural functions used to generating GCM data has been provided in the main paper and Section F. For the synthetic data beyond GCM, $U$ is drawn from a uniform distribution on $[0, 1]$. $A$ is a binary attribute with equal probability. $U_0$ is sampled from the uniform distribution of the set $\{1, ..., 8\}$ and translated into a one-hot vector.

For the law school admission dataset, race is a categorical attribute with 8 classes. So we translated it into one-hot vector. In the next paragraphs, we did the same operation for $X_0$ and Race, $A$ and Sex, and $X_1$ and GPA, $X_2$ and LSAT.

We concatenated $[A, X_0, X_1, X_2]$ as the representation for UF baseline. A LinearRegression model provided by sklearn package was used to predict $Y$. For testing A-Accuracy, we used an SVM classifier. For our CF method, we used the same prediction model and same SVM classifier. The input representation was obtained by concatenating $[U, X_0, \frac{X_1 + \check{X}_1}{2}, \frac{X_2 + \check{X}_2}{2}]$.

Baselines are also tested using the same method after obtaining the fair representation. For the CI baseline, we used the same architecture as Oh et al. (2022). To fit the regression task, we replaced loss function of the target network with a MSE Loss. The hyper-parameter $\alpha$ was set as 4.0. For MaxEnt-ARL baseline, we used the same architecture and set the hyparameter $\alpha$ as 4.0. Because it is hard to get a fair representation than the dataset used in Oh et al. (2022), we updated the discriminator for 10 steps in every training iteration. We also used the architecture prvided by Oh et al. (2022) to train the FarconVAE-t and FarconVAE-G model. Their model contains a encoder and decoder. We changed the construction error into two parts, cross entropy loss for constructing $X_0$ and MSE Loss for constructing $X_1$ and $X_2$. For FarconVAE-t baseline, we set $\alpha = 2.0, \beta = 0.15, \gamma = 0.75$. For FarconVAE-G baseline, we set $\alpha = 1.0, \beta = 0.05, \gamma = 1.0$.

# I  Reproduce the Result

Directory GCM_Simulation contains the code for the experiment with synthetic data generated by GCM. To get the result in Table .1, run the following command:

```
1  cd GCM\_Simulation
2  python main.py
```

Directory Non_GCM_Simulation contains the code for the experiment with synthetic data generated beyond GCM. To get the result in Table 2, run the following command:

```
1   cd Non_GCM_Simulation
2   # generate representations
3
4   # FarconVAE-t baseline
5   CUDA_VISIBLE_DEVICES=0 python main.py --scheduler=one --kernel=t --alpha=2.0 --beta=0.15
        --gamma=0.75 --model_name FarconVAE-t
6
7   # FarconVAE-G baseline
8   CUDA_VISIBLE_DEVICES=0 python main.py --scheduler=one  --kernel=g --alpha=1.0 --beta
        =0.05 --gamma=1.0 --model_name FarconVAE-G
9
10  # MaxEnt-ARL baseline
11  CUDA_VISIBLE_DEVICES=0 python main_maxent.py --scheduler=one --alpha=4.0 --model_name
        MaxEnt-ARL
12
13  # CI baseline
14  CUDA_VISIBLE_DEVICES=0 python main_maxent.py --scheduler=one --alpha=4.0 --model_name CI
15
16  # UF baseline
17  python main_uf.py
18
19  # our method
20  python main_cf.py
21
22  # evaluate representations
23  python evaluate.py
```

Directory Law contains the code for the experiment with Law School Admission dataset. To get the result in Table 3, run the following command:

```
1  cd Law
2  # generate representations
3
4  # FarconVAE-t baseline
5  CUDA_VISIBLE_DEVICES=0 python main.py --scheduler=one --kernel=t --alpha=2.0 --beta=0.15
        --gamma=0.75 --model_name FarconVAE-t
6
7  # FarconVAE-G baseline
8  CUDA_VISIBLE_DEVICES=0 python main.py --scheduler=one  --kernel=g --alpha=1.0 --beta
        =0.05 --gamma=1.0 --model_name FarconVAE-G
```

```
 9
10   # MaxEnt-ARL baseline
11   CUDA_VISIBLE_DEVICES=0 python main_maxent.py --scheduler=one --alpha=4.0 --model_name
         MaxEnt-ARL
12
13   # CI baseline
14   CUDA_VISIBLE_DEVICES=0 python main_maxent.py --scheduler=one --alpha=4.0 --model_name CI
15
16   # UF baseline
17   python main_uf.py
18
19   # our method
20   python main_cf.py
21
22   # evaluate representations
23   python evaluate.py
```

## J   Related Work

With the development of machine learning models, fairness, as an important potential risk, has been studied in many previous works (Xie et al., 2024a;b; Xie & Zhang, 2024; Du et al., 2020). Suppose the machine learning task is built on the distribution $(A, X, Y)$, in which $A$ represents the sensitive attribute determined by social norms. $X$ is the set of observed features other than the sensitive attribute, and $Y$ is the target attribute. We use $\hat{Y} = g_y(X, A)$ to represent the prediction of $Y$ by the machine learning model $g_y$.

Fairness through unawareness (Calders et al., 2009) is a fairness definition that regards $\hat{Y} = g(X)$ as fair, implying that omitting the sensitive attribute $A$ from the model ensures fairness. Statistical parity (Besse et al., 2022), also referred to as demographic parity, is achieved when there is independence between $\hat{Y}$ and $A$. Conditional statistical parity (Corbett-Davies et al., 2017) is a relaxation of this independence, applying the requirement to a subset of data instances. Equalized odds Romano et al. (2020), based on another kind of statistical independence called separation, is satisfied when a classifier has identical true positive rates and false positive rates across different protected groups. Equal opportunity (Wang et al., 2019) is a relaxed version of separation, requiring only the same false negative rate among groups. Sufficiency, the basis for the fairness definition known as calibration (Salvador et al., 2021), is met when instances with the same prediction have the same likelihood of belonging to the positive class. This concept can also be relaxed to predictive parity (Zeng et al., 2022), which requires the classifier to maintain the same positive predictive value across different groups.

To achieve statistical fairness, three main types of methods have been extensively studied. Pre-processing methods are applied directly to the data to foster the development of fair AI models (Zhang et al., 2022). A common approach in this category is reweighting the data, which typically involves a three-step process: massaging the original labels, reweighting, and resampling (Kamiran & Calders, 2012). These steps collectively aim to adjust the data distribution to mitigate inherent biases, setting the stage for more equitable model training and outcomes.

Post-processing methods usually involve modifying the predictions of an existing model. For example, Petersen et al. (2021) introduced a technique involving graph smoothing applied to the output of an NLP model to achieve individual fairness. Kim et al. (2019) proposed a black-box approach for post-processing results, while Lohia (2021) used a priority-based method to simultaneously achieve group and individual fairness. A key advantage of post-processing methods is their ease of implementation across different models without requiring retraining. However, these methods often present design challenges and are typically tailored for specific fairness objectives, limiting their general applicability.

In-processing methods involve applying fairness constraints during model training. Learning a fair representation is one of the most common approaches. To remove sensitive information from the representation, techniques like adversarial learning (Feng et al., 2019) and disentanglement (Locatello et al., 2019) are often employed. Controllable-invariance (CI) (Xie et al., 2017) includes an encoder, a discriminator, and a predictor, using a minmax game to make the representation invariant to the sensitive attribute. Maximum Entropy Adversarial Representation Learning (MaxEnt-ARL) (Roy & Boddeti, 2019) addresses the sub-

optimal problem. Disentanglement often utilizes a Variational Autoencoder (VAE) to apply constraints on the latent space. Variational Fair Autoencoder (FVAE) (Louizos et al., 2015) minimizes the maximum mean discrepancy (MMD) on the posterior distributions. Orthogonal Disentangled Fair Representations (ODFR) (Sarhan et al., 2020) forces sensitive and non-sensitive representations to be orthogonal. Fair representation via Distributional Contrastive Variational AutoEncoder (FarconVAE) (Oh et al., 2022) employs contrastive learning to reduce correlation in the representation space.

Individual fairness emphasizes the fairness property on individual data points, requiring similar prediction on similar data pairs (Dwork et al., 2012). The basic method to achieve IF is to solve an optimization under the IF constraints. Ifair (Lahoti et al., 2019) added fairness regularizer to the basic objective functions. Post-processing method can also be used. GLIF (Petersen et al., 2021) reframed the post processing step as a graph smoothing problem, which is computationally efficient. Beyond the observed distribution $(A, X, Y)$, counterfactual fairness leverages the structural causal model (SCM) $\mathcal{M}(U, V, F)$ underlying the data (Pearl, 2010).

Kusner et al. (2017) proposed a definition of counterfactual fairness as the equality of predictive distribution in both factual and counterfactual worlds. They also provided a method to achieve this fairness by utilizing only the exogenous variables $U$. Building on this approach, Zuo et al. (2023) extended counterfactual fair representation by employing a symmetric function.

Generally speaking, counterfactual fairness and parity-based fairness are not equivalent (Silva, 2024). Rosenblatt & Witter (2023) attempted to bridge the gap between counterfactual fairness and demographic parity. However, their work used a stronger assumption than the definition of counterfactual fairness. Furthermore, Anthis & Veitch (2023) demonstrated, in a special case where no exogenous variable exists, how counterfactual fairness and group fairness can be interconnected.

## K   Computation Resources

When doing the simulation and experiments for the paper, we used a server with 64 CPUs. The model name of the CPUs is AMD EPYC 7313 16-Core Processor. The server has 8 RTX A5000 GPUs, with 24GB memory for each one. For the experiment, we used only one single GPU. The experiment on synthetic data takes less than one hour for each independent run. The experiment need around 6 hours on Law School Admission data.

