# OpenReview forum: "On the Connection Between Counterfactual Fairness, Statistical Parity and Individual Fairness"
_TMLR — Rejected by TMLR_

### Review · Reviewer_6cwE · 2024-12-13

**Summary Of Contributions:**

The paper studies the relation between counterfactual fairness--a causal notion of fairness--and non-causal notions, such as statistical parity and individual fairness. A piece of prior work has proposed building predictors based on counterfactually fair representations as a way to implement counterfactual fairness in practice. The authors identify theoretical conditions that the underlying causal model (i.e., the data-generating process) must satisfy, so that using counterfactually fair representations also implies that individual fairness and/or statistical parity are satisfied. Moreover, they illustrate their theoretical findings with a series of experiments using both synthetic and real data and comparing with several baselines from the fairness literature.

**Audience:**

Yes

**Broader Impact Concerns:**

The authors already include a "Limitations and Social Impact" section, but it is rather short. I think this part could be further expanded with a clear discussion of the limitations of the work:

(i) pointing out that the paper has focused on learning counterfactually fair representations and not on abstract connections between counterfactual fairness and other non-causal notions of fairness

(ii) highlighting that one needs to know the underlying SCM to impose counterfactual fairness to the predictor and clarifying that there could be negative social impact if there is a mismatch between the true and the assumed SCM

(iii) discussing how restrictive are the assumptions made about the SCM for the results of section 3 to hold, potentially with a few examples of situations where these assumptions may be true in practice

**Claims And Evidence:**

No

**Requested Changes:**

In my opinion, there are several changes that the authors can do, which would improve the paper and would help them highlight the contributions of their work. I present my suggestions organized along three axes, based on the aforementioned weaknesses.

*Framing and connection with related work*:
* The paper's title and multiple parts of the text make claims about the connection between counterfactual fairness, statistical parity and individual fairness. However, the technical part of the work focuses on counterfactually fair representations, which is a specific way to *implement* counterfactual fairness, proposed by Zuo et al. (2023). Therefore, I believe that the scope of the paper is somewhat narrower than it originally appears to be. When the authors first mention the work by Zuo et al., they write that *"it is the most general way to guarantee counterfactual fairness to our best knowledge."* However, there are no further arguments provided and this is taken as a main assumption for the rest of the paper. I believe that, if the authors want to make general claims about counterfactual fairness, they should provide technical results based solely on the original definition by Kusner et al. (2017), otherwise they should reframe their paper to better reflect its true scope and its focus on counterfactually fair representations.
* Although the literature on the connections between counterfactual fairness and other non-causal notions of fairness is relatively sparse, there are some works that have already explored that research direction. Importantly, the works by Rosenblatt & Witter (2023) and Silva (2024) seem to make related (and even contradicting) claims to the current paper, but there is no in depth discussion of the similarities and differences, only a brief discussion at the end of page 6. I believe that clearly differentiating the current paper with those two works is crucial and it should be done in the introduction of the paper much more thoroughly.
* The paper seems to be missing a broader discussion of other works at the intersection of fairness and causality. Here are some indicative citations for reference [1, 2, 3].

*Formulation and analysis*:
* The authors use the framework of structural causal models (SCMs) to formalize their setting. They provide an accurate general definition of SCMs in page 2, however, the way they treat SCMs in section 3 is a bit problematic. One central aspect of an SCM is that one can capture all the randomness in a system using exogenous (noise) variables $U$ and then express all endogenous random variables in the system as a function (i.e., deterministic mechanism) of their ancestors. The authors introduce a Gaussian causal model (definition 2) which includes exogenous variables $U$ sampled from a Gaussian distribution, yet, the endogenous variable $X$ is still characterized by a distribution and is not given as a function of its ancestors. I think this can be easily solved if the authors introduce an additional exogenous variable $U_x \sim \text{Gaussian}(0, \Sigma_x^2)$ to capture that randomness and write $X = W_u U + f_a(A) + b + U_x$, however, the current formulation is non-standard and may cause confusion.
* The paper is missing a clear definition and explanation of what a counterfactually fair representation is, although this is a central concept mentioned and used throughout the paper. Importantly, it is unclear if the true SCM needs to be fully known in order to build those representations. This is an important point and it should be clarified in the paper. Moreover, the authors introduce two fundamentally different definitions in page 4. First, they introduce counterfactually fair representations as $H(x,a)$, which is a random variable. Importantly, the expectations used for the definition do not specify what is the source of randomness, making the definition hard to follow. Then, they introduce $r(x,a)$ as a deterministic version of counterfactually fair representations. I think it would be very helpful for the clarity and consistency of the paper if the authors decided on *one* definition of counterfactually fair representations and built the rest of the paper based on that. It is quite hard to understand their results without this point being properly clarified.
* For Theorem 1, the authors assume that the function $f_a$ is Lipschitz continuous. However, $f_a$ seems to take as input the sensitive attribute $A$ which, according to footnote 2 (and naturally), takes a finite number of values. It is unclear what Lipschitz continuity means in that context, and I think it would be helpful if the authors specified what is the domain of $f_a$ and wrote their assumption formally.
* In page 5, the authors provide an example of a causal model where an optimal predictor can satisfy statistical parity without being counterfactually fair. I would encourage the authors to expand and discuss the details of the example step-by-step. Currently, it is hard to follow. Specifically, the inequality $||\hat{Y}(x,a)-\hat{Y}(x',a')|| \leq ...$ seems to appear out of nowhere.
* Similar to the previous point about Lipschitz continuity, in section 3.3., the authors make the assumptions that $f_a$ is linear. It is unclear what linearity means for a function defined over a finite set, and the authors should also try to write this assumption formally.
* The authors' use of the term "uniform distribution" is ambiguous and makes the paper hard to follow, as well as its proofs. For example, in section 3.3, they mention that $A$ follows a uniform distribution where $A$ is (presumably) a discrete random variable. However, in section 3.4, they write that $U$ follows a uniform distribution, where $U$ is (presumably) a continuous random variable. In the proof of theorem 3 (eq. 68), the authors seem to use the fact that $P_A(a)$ is a uniform distribution to imply $p_A(a_1)=1$, which would be the case for continuous random variables but not for discrete ones. I think those details should be clarified in the paper and its proofs, so that there is no confusion.

*Organization of the paper*:
* Section 2 is called "problem formulation", but it includes mostly general definitions of SCMs and notions of fairness from prior work. If they want to introduce these in the main body of the paper, I would encourage the authors to introduce them as "Preliminaries". Section 2 does not contain any concrete problem formulation--there are two high-level questions that the paper is trying to answer, but these could be better placed in the introduction.
* Section 3 is called "Theoretical result", however, section 3.1 is not a result. It is basically a relatively short introduction of counterfactually fair representations, as they were introduced by Zuo et al (2023). As I mentioned earlier, this part of the text could be improved, and it would also be a better fit either in the introduction or in the preliminaries.
* I found the last part of the experiments (section 4.4) slightly confusing, and I believe that it doesn't add much in comparison with the rest of section 4. Specifically, the authors work with a face recognition dataset, and they treat the light conditions in each image as a sensitive attribute. First of all, this experimental setup is unrealistic, since the choice of light conditions in an image recognition task seems to be very far from what other papers in the area consider as a sensitive attribute (e.g., race, gender, age). In addition, this choice seems to lead to experimental results that are fundamentally different from the other subsections. For example, in Table 4, the predictor that is not optimized to satisfy any notion of fairness seems to perform the worst in terms of accuracy, while fair predictors perform better. This is in contrast to the results of previous sections, where there is a trade-off between maximizing accuracy and maintaining different notions of fairness, including counterfactual fairness. That said, I would encourage the authors to either better justify their experimental setup in 4.4 and discuss the differences with previous subsections or just skip this subsection altogether.

[1] Plecko, Drago, and Elias Bareinboim. "Reconciling predictive and statistical parity: A causal approach." Proceedings of the AAAI Conference on Artificial Intelligence. Vol. 38. No. 13. 2024.

[2] Nilforoshan, Hamed, et al. "Causal conceptions of fairness and their consequences." International Conference on Machine Learning. PMLR, 2022.

[3] Makhlouf, Karima, Sami Zhioua, and Catuscia Palamidessi. "Survey on causal-based machine learning fairness notions." arXiv preprint arXiv:2010.09553 (2020).

**Strengths And Weaknesses:**

The **main strength** of the paper is its focus on an important topic in the fairness literature, namely, the connection of counterfactual fairness to other notions of fairness. Although there is a significant amount of work studying the trade-offs and conflicts between different non-causal notions of fairness, to the best of my knowledge, there have not been many such findings regarding counterfactual fairness. The authors have done a good job motivating the problem in the introduction, and they have done a lot of analysis for this paper, both theoretical and experimental. Therefore, the paper could be a valuable addition to the literature and attract significant interest in the TMLR community.

However, the paper presents 3 **weaknesses** that I consider important, and I briefly list below. Under "Requested Changes", I provide some actionable feedback that may help to address them.
* The work is not properly contextualized with respect to the closely related literature.
* Some parts of the mathematical formulation/analysis need to be handled with better care.
* The organization of the main content of the paper can be further improved.

---

> ### Author Response · Authors · 2025-01-02
>
> **Forming connection with related work**
>
> * Point 1: Our work focuses on the connection between counterfactual fairness and other fairness notions when counterfactual fairness is achieved through counterfactually fair representation. Thank you for your comment. We revised the title, abstract, introduction and other section to clarify this point.
>
> * Point 2: The difference between our paper and Rosenblatt & Witter (2023)/Silva (2024) is as follows. Rosenblatt & Witter (2023) show that when $A \perp U$, counterfactual fairness and statistical parity can imply each. However, they only consider the case when the predictor is a function of $U$. While thier proof technique might be useful to show that representation $H(X, A)$  implies statistical parity, we cannot use their results and technique to show that r(X,A) leads to statistical parity. Silva (2024) also pointed out that the proof in Rosenblatt & Witter (2023) is not comprehensive, but Silva (2024) did not discuss when CF implies SP.
>
> * Point 3: Thank you for mentioning these works. We added them in the introduction of the revised edition.
>
> **Formulation and analysis**
>
> * Point 1: Non-deterministic SCMs have also been stated and used in the counterfactual fairness paper [Kusner et al. 2017] (see Section 4.2). The full deterministic model requires stronger assumptions. As stated in [Kusner et al. 2017], while deterministic ($X = W_{u}U + f_{a}(A) + b + U_{x}$) and non-deterministic ($X \sim \mathrm{Gaussian}(W_{u}U + f_{a}(A) + b, \Sigma_{x}^{2})$) SCMs can lead to the same data distribution, the counterfactual reasoning under these two models can be different, and we should use the one that expresses our scenario better. For example, in the law school admission (see Section 5 in Kusner et al. 2017), we may want to consider knowledge as the only unobserved variable which is an important factor for fairness in admission process. In this case, the causal model should be expressed in a non-deterministic form. Note that the non-deterministic SCMs are also widely considered in the literature [Chiappa 2019] (see Section titled PSCF-VAE), [Grari et al. 2023] (see Section 4) and [Abroshan et al. 2022].
>
> * Point 2: A counterfactually fair representation is a representation which has the same distribution in the factual world and counterfactual world. We provided a formal definition (Definition 1) in Section 2.2 in the revised edition.
>
>     The randomness of the expectation in the definition of $H(x, a)$ comes from the randomness in the structural functions in non-deterministic causal model (e.g., gaussian causal model). When the structural functions are deterministic, for example, $X = f(U, A)$, $H(x, a)$ should be $[s(f(U, \check{a}^{[1]}), ...), U]$.
>
>     We want to use two kinds of counterfactually fair representations because the second one (i.e., r(x,a)) is neglected in previous work (Rosenblatt & Witter (2023)), resulting in an incomprehensive conclusion. In particular, the proof technique and results in (Rosenblatt & Witter (2023)) cannot be applied to r(x,a). Therefore, we cannot directly extend the results for representation H(x,a) to representation r(x,a).  In our paper, we consider both cases separately for every theorem.
>
> * Point 3: The domain of $A$ can be discrete or continuous. The metric space for $f_{a}(a)$ is the L2 norm. That is, the Lipschitz continuous assumption for $f_{a}$ implies that $||f_{a}(a’) - f_{a}(a)|| \leq L_{a} ||a’ - a||$ (here $||\cdot||$ means the L2 norm). We added this assumption clearly in the revised edition.
>
> * Point 4: The intuition for example 1 (example 2 in the revised version) is that from Kusner et al. 2017, we know that a counterfactually fair predictor cannot take sensitive attribute $A$ or any descendent of $A$ as input. On other hand, it is possible to create an individually fair predictor that takes the sensitive attribute as input. In this example, we identified an individually fair predictor that takes the sensitive attribute as input. As a result, it cannot be counterfactually fair, and we showed that mathematically. We also provide a step-by-step illustration for the example in the revised edition in Appendix E.
>
> * Point 5: $f_{a}(a)$ is linear means $f_{a}(a) = W_{a}a$, no matter whether $A$ is discrete or continuous. We added the formal definition in the revised edition.
>
> * Point 6: In Eq.68, we are using the condition that $p_{A}(a_{1}) = p_{A}(a_{2})$, not $p_{A}(a_{1}) = 1$. For discrete distribution, uniform distribution refers to $p_{A}(a_{1}) = ... = p_{A}(a_{\mathcal{A}})$. For continuous distribution, uniform distribution means $p_{U}(u) = \frac{1}{|\mathcal{U}|}$, where $|\mathcal{U}|$ is the volume of the domain of $U$. We made it clear in the revised edition.

---

> ### Author Response · Authors · 2025-01-02
>
> **Organization of the paper**
>
> * Point 1: Thank you for the suggestion. We stated it as a Preliminaries section.
>
> * Point 2: We moved Section 3.1 to the preliminaries section (now section 2.2).
>
> * Point 3: For experiment 4.4, our goal is to consider a scenario where the causal model is not known and study whether CFR implies SP. However, since you did not find it valuable, we removed it in the revised version. Please let us know if you want us to add another experiment. We will add another experiment based on your feedback.
>
> **Border impact concerns**
>
> * Thank you for your suggestions. We improved our discussion in the revised edition.

---

> > ### Comment · Reviewer_6cwE · 2025-01-06
> >
> > I would like to thank the authors for their response and for incorporating my suggestions in the edits they did to the original version of the manuscript. My main concerns regarding the scope of the paper, its connection to prior work and the organization of sections have been addressed.
> >
> > However, I still think there are (some) aspects of the formulation that require further clarification. Regarding my original points:
> >
> > - Point 1: I still don't understand why non-deterministic SCMs are introduced and how they work. If there is randomness in the generating process, there has to be some variable, even if that is considered unobservable, which introduces that randomness. I agree that, to model a generating process $X\sim \text{Gaussian}(W_uU+f_a(A)+b, \Sigma_x^2)$ as an SCM, the structural equation does not necessarily have to be $X = W_uU+f_a(A)+b + U_x$ as I suggested, but it has to take the form $X = g(U, A, U_x)$ for *some* function $g$ and *some* distribution of $U_x$, such that the $X$ follows a Gaussian distribution. Note that, the former would lead to deterministic counterfactuals for $X$, while the latter would lead to a distribution of counterfactual values. In both cases, the structural equation itself is deterministic. Moreover, when SCMs are first introduced in the preliminaries of the paper, they are introduced using *structural equations*, not with the form adopted in Section 3. The authors explicitly write "Since any observable variable is fully determined by unobserved variables $U$ and structural equations, the counterfactual value of $Z$ given $U = u$ can be computed by replacing $U$ with the value $u$ in structural equations and replacing structural equation for $Q$ by $Q = q$." However, when they introduce the Guassian causal model in Section 3, $X$ is not fully determined by the unobserved variables $U$ (since there is still randomness after conditioning on $U$) and there is no structural equation, so the whole process of computing counterfactuals in this model does not seem very clear/formal.
> >
> > - Point 3: To say that the function $f_a$ is Lipschitz-continuous, one needs to specify a metric space, that is, a set (i.e., the domain of $f_a$) and a distance function between the elements of the set. My understanding is that, in the paper, the domain of $f_a$ is $A$, the set of sensitive attribute values. However, according to footnote 2, that set is finite. It is not clear to me how the distance function can be the L2 norm in that case, since norms are usually defined over vector spaces. I think the authors need to clarify what are the values of $A$, that is, if they are discrete elements (i.e., demographic groups) or some kind of vector representations (of demographic groups).
> >
> > - Point 5: Similar to the previous point, the authors mention that $f_a$ is linear, that is, $f_a(a)=W_a a$, no matter whether $a$ is discrete or continuous. However, this assumption implies that $a\in A$ is a vector, otherwise multiplying with a matrix $W_a$ is not well-defined. Therefore, I don't think $a$ can be discrete---although it seems to be according to footnote 2. Since the main theorems of the paper rely on linearity and Lipschitz continuity, I think the authors should revisit these assumptions and clearly explain what the set $A$ is. Currently, the math is a bit confusing.
> >
> > - Point 6: Regarding Eq. 68 (67 in the revised version), I believe the use of the uniform distribution is fine, but there seems to be a typo. I think $p_{r_x, r_u \mid A} (r_{x0}, r_{u0}\mid a_1) = \frac{p_{r_x, r_u \mid A} (r_{x0}, r_{u0}\mid a_1)}{p_A(a_1)}$ should be $p_{r_x, r_u \mid A} (r_{x0}, r_{u0}\mid a_1) = \frac{p_{r_x, r_u, A} (r_{x0}, r_{u0}, a_1)}{p_A(a_1)}$, otherwise it would hold that $p_A(a_1)=1$.

---

> > > ### Author Response · Authors · 2025-01-07
> > >
> > > * Point 1: We would like to quote the following sentence from Kusner et al. 2017, “we point out that we do not need to specify a fully deterministic model, and structural equations can be relaxed as conditional distributions. In particular, the concept of counterfactual fairness holds under three levels of assumptions of increasing strength.”
> > >
> > >    As a result, since counterfactual fairness definition is valid in non-deterministic causal models, we tried to consider this type of causal model in our paper as well. We agree that we did not provide the definition for non-deterministic causal model in our preliminaries. We will modify Section 2.1 to include a clear definition of non-deterministic causal models. This can be done by adding a few sentences.
> > >
> > >    We also want to clarify that in non-deterministic gaussian causal model, finding $H(x,a)$ is very straightforward. In particular, random variable $H(x,a)$ under non-deterministic Gaussian causal model is given by $[s(W_{u}U + f_{a}(a^{[1]}) + b, ...,  W_{u}U + f_{a}(a^{[|\mathcal{A}|]}) + b), U]$ where $U$ follows the conditional distribution given $X = x, A = a$.
> > >
> > > * Point 3: $A$ could be discrete variables (such as categorized financial status poor, good, etc.) or continuous variables (such as credit scores). Note that $||a – a'||_{2} = \sqrt{(a – a')^{2}}$ and it is consistent with the definition of $n$-dimensional vectors when $n = 1$. Thank you for pointing out that this is not standard, and we will modify it in the revised edition.
> > >
> > > * Point 5: $f_{a}(A)$ is a scalar variable here. $W_{u}U + f_{a}(A)$ here is a vector which $f_{a}(A)$ will be added to each dimension of $W_{u}U$. We will explain the math clearly in the revised edition.
> > >
> > > * Point 6: Thank you for pointing out our typo here. We will correct it in the revised edition.

---

### Review · Reviewer_gt7b · 2024-12-14

**Summary Of Contributions:**

The paper studies the relationship of counterfactually fair representations (representations that allow a model trained on them to satisfy counterfactual fairness) to two observational notion of fairness, individual fairness and statistical parity. In particular, they characterize two types of causal models where under certain assumptions where counterfactually fair representations implies the latter notions of fairness. They also show that under the same assumptions individual fairness and statistical parity does not imply counterfactual fairness.

**Audience:**

Yes

**Broader Impact Concerns:**

I do not think a Broader Impact Statement is necessary.

**Claims And Evidence:**

Yes

**Requested Changes:**

- I think it is confusing at some points to that the results are presented as connecting counterfactual fairness and SP/IF, but the shown implication is only counterfactual fair representations implying SP/IF
- I find experiment 4.4 rather odd by assuming the light condition as sensitive attributes and taking the observational data as counterfactuals. It does not add anything new to the other experiments and is rather confusing or missing some context (e.g., best A-Accuracy being 20% not 50% anymore and not really discussed in the text, unclear why baseline performs worse than the fair models where this trend was not observed before). I think the experiment needs more context or should be removed from the main text.
- Section 3 should be revised. There are multiple mistakes and unclear sentences in the text that could be improved upon. For example:
 unclear last sentence in paragraph before section 3.2.; for "On the other hand, in GCMs, individual fairness does not necessarily imply counterfactual fairness." helpful to add "under the same conditions" as in general both implications are untrue; for "Our
results are consistent with Silva (2024) as we show that under assumptions A ≃ U , counterfactual fairness
and statistical parity can not imply each other.", you only show that SP does not imply CF (under conditions of Theorem 4).
- An intuition for example 1 would be nice.

**Strengths And Weaknesses:**

Strengths
- The work is well motivated in the introduction.
- I think the results are interesting in a theoretical sense though not conveyed and discussed in a friendly way as to understand the implications to using counterfactual fair representations in real world applications (can the assumptions made be tested, etc.).

Weaknesses
- The use of non-deterministic SCMs is rather unusual and quite limiting for the theoretical results of theorem 3, and the assumption of a uniform noise distribution P(U) seems also quite limiting for theorem 4. In that regard, a discussion of these assumptions and how it affects the application of these results is missing, the current limitation section is not really informative.
- It is unclear over what metric space the Lipschitz assumption for function f_a is over since \Acal is categorical, this should be clarified as it is a crucial assumption for Theorems 1-3

---

> ### Author Response · Authors · 2025-01-02
>
> * Weaknesses 1: Non-deterministic SCMs have also been stated and used in the counterfactual fairness paper [Kusner et al. 2017] (see Section 4.2). The full deterministic model requires stronger assumptions. When we do not have such knowledge, the structural equations can be relaxed as conditional distributions. This kind of SCMs is also widely considered in the following work, such as [Chiappa 2019] (see Section titled PSCF-VAE), [Grari et al. 2023] (see Section 4) and [Abroshan et al. 2022].
>
>    Regarding distribution of $U$, the distribution of $U$ depends on the causal model that we consider. Since normal or uniform distributions are commonly used in literature, we focused on these distributions. Generally speaking, for other types of distribution, we need separate study.
>
>    Regarding the assumptions, we revised the limitation section to discuss the underlying assumptions in our paper.
>
> * Weakness 2: The metric space for $f_{a}(a)$ is the L2 norm. That is, the Lipschitz continuous assumption for $f_{a}$ implies that $||f_{a}(a) - f_{a}(a')|| \leq L_{a}||a' - a||$ (here $||\cdot||$ is the L2 norm). We added this assumption clearly in the revised edition.
>
> * Requested changes 1: Our work focuses on the connection between counterfactual fairness and other fairness notions when counterfactual fairness is achieved through counterfactually fair representation. Thank you for your comment. We revised the title, abstract, introduction and other section to clarify this point.
>
> * Requested changes 2: For experiment 4.4, our goal is to consider a scenario where the causal model is not known and study whether CFR implies SP. However, since you did not find it valuable, we removed it in the revised version. Please let us know if you want us to add another experiment. We will add another experiment based on your feedback.
>
> * Requested changes 3: We revised section 3 based on your feedback. Example 4 in Appendix D is an example to show that SP does not necessarily implies CF.
>
> * Requested changes 4: Intuition for example 1 (example 2 in the revised version): from Kusner et al. 2017, we know that a counterfactually fair predictor cannot take sensitive attribute $A$ or any descendent of $A$ as input. On the other hand, it is possible to create an individually fair predictor that takes the sensitive attribute as input. In this example, we identified an individually fair predictor that takes the sensitive attribute as input. As a result, it cannot be counterfactually fair, and we showed that mathematically.
>
> 1. Kusner, Matt J., et al. “Counterfactual fairness.” Advances in neural information processing systems 30 (2017).
> 2. Chiappa, Silvia. “Path-specific counterfactual fairness.” Proceedings of the AAAI conference on artificial intelligence. Vol. 33 No. 01. 2019.
> 3. Grari, Vincent, Sylvain Lamprier, and Marcin Detyniecki. “Adversarial learning for counterfactual fairness.” Machine Learning 112.3 (2023): 741-763.
> 4. Abroshan, Mahed, Mohammad Mahdi Khalili, and Andrew Elliott. “Counterfactual fairness in synthetic data generation.” NeurIPS Workshop on Synthetic Data for Empowering ML Research. 2022

---

> > ### Comment · Reviewer_gt7b · 2025-01-08
> > **Follow up on weakness 1 and 2**
> >
> > I thank the authors for their response and the changes made to address some of my concerns.
> > However, I still think that weakness 1 and 2 requires further clarification:
> >
> > Weakness 1: I took a look at the 4 papers that the authors cite in the comment that consider non-deterministic SCMs. From the four papers, I only found Abroshan et al. [2022] to use non-deterministic SCMs in the paper. They learn it as part of the experiment section but never formalize it either. Kusner et al. [2017] only mentions the possibility of non-deterministic SCMs. Chiappa [2019] uses graphical causal models (GCMs) not structural causal models. Finally, in Grari et al. [2023] they do not introduce or talk about structural causal models at all, but seem to also only use a graphical causal model.  To clarify, I am not implying that the models itself are wrong, there is work using non-deterministic graphical causal models as pointed out by the authors, but there is no proper formalization of non-deterministic SCMs in prior works. I believe this is because SCMs were introduced as a way to account for randomness in a causal model through unknown exogenous variables. Hence, if you fix these variables there should be no randomness in your model. Formalizing non-deterministic SCMs defeats this purpose and is therefore rather unusual. I don't understand why the authors didn't use graphical causal models instead.
> >
> > Weakness 2: I have also read the authors response to reviewer 6cwE related to this point and having a L2 norm over a categorical space doesn't make a lot of sense to me. For example, if the sensitive attribute is race, how would one measure a L2 norm over this unordered set. One would need to represent $a$ as some kind of vector, and even if we can compute it, what would be the interpretation of a distance between races. I don't think this makes sense for all types of sensitive attributes and the authors should be more precise in their problem formalization.

---

> > > ### Author Response · Authors · 2025-01-10
> > >
> > > * Weakness 1: Thank you for your suggestion. Kusner et al. 2017 also used non-deterministic causal model in their experiments. We agree that graphical causal model is a better term to describe gaussian causal model. If you want us to reframe Section 3.1 and 3.2 with graphical causal model, we are very glad to do that.
> > >
> > > * Weakness 2: The L2 norm metrics might not be suitable for all cases of sensitive attributes. The most suitable case is when $A$ is a continuous variable, such as Age. When $A$ is discrete, if it is a binary an ordered multi-value feature, such as income level, it is still meaningful for using the distance $|a – a'|$. We have made more explanation and discussion in the revised edition.

---

### Review · Reviewer_dZHZ · 2024-12-24

**Summary Of Contributions:**

The authors study the connections between counterfactual fairness and two observational fairness notions - statistical parity and individual fairness. They provide a set of assumptions under which the counterfactually fair representation method given in Zuo et al (2023) guarantees both observational fairness notions. Specifically, they consider a form of causal model they describe as a Gaussian causal model and demonstrate that in this case counterfactual fairness implies both notions, as well as providing a separate theorem which demonstrates that counterfactual fairness implies statistical parity more broadly. Finally, they include a selection of  experiments which demonstrate that the counterfactually fair representation method can also give good predictive performance whilst satisfying the studied observational forms of fairness.

**Audience:**

Yes

**Claims And Evidence:**

No

**Requested Changes:**

1. Can the authors explain to me what is wrong with the argument I have presented in weakness 2? Specifically, why can't $H(X,A)$ be written as $(f(U),U)$ for $U$ is sampled from $P(U|X=x,A=a)$?
2. If no arguments can be presented against my weaknesses 2 and 3 above, I suggest that the reviewers remove the links between parity and counterfactual fairness all together, as well as anything on $r(X,A)$.
3. Changes should also be made to include the counterfactual invariance literature, and position their work in relation to that.
4. What can be guaranteed by Zuo et al (2023) should be toned down to reflect the fact that knowledge of the true SCM is required.

**Strengths And Weaknesses:**

Strengths:
1. I believe the links between counterfactual fairness and individual fairness are novel and interesting. I have not seen anything along these lines before and it seems like a worthy area of study.

Weaknesses:
1. The authors seem to have missed quite a large amount of work that contains links between counterfactual fairness and statistical parity, specifically the area of counterfactual invariance. In the literature on counterfactual invariance there are numerous works which contain results on this relationship [1,2,3] as well as other works in fairness [4]. Moreover, even the cited work [5] seems to already cover more general versions of the results on counterfactual fairness and statistical parity in the paper (see next point).
2. I do not understand why the results on counterfactual fairness and statistical parity given in Rosenblatt & Witter (2023) don't immediately imply more general versions of the results given by authors in section 3.3 and 3.4. The authors state that this is because they consider the counterfactually fair representation of  Zuo et al (2023), whereas Rosenblatt & Witter (2023) only consider functions of $U$, however it would seem to me that the counterfactually fair representation is also a function of $U$ just with more steps. Specifically the term $s\left(\mathbb{E}\left[\check{X}\left[\check{a}^{[1]}\right] \mid U\right], \ldots, \mathbb{E}\left[\check{X}\left[\check{a}^{|\mathcal{A}|}\right] \mid U\right]\right)$ only takes as input $U$ and so $H(x,a) = (f(U),U)$ for some function $f$ where $U$ is sampled from $P(U|X=x,A=a)$. Given this the  $A \perp U$ would be sufficient to imply $H(X,A) \perp A$ without any additional assumptions or specific causal models.
3. This leads into the example in Appendix D which shows that $A \perp U$ does not imply $A \perp r(X,A)$. It appears to me that this same example can be used to show that $r(x,a)$ is not counterfactually fair. This follows from the fact that we have:
$$ P( r(X,A) = 0.3 | X= 0, A=-1) = 1$$
But subbing in the counterfactual so that a= 1 we have:
$$ P( r(X(a),a) = 0.3 | X= 0, A=-1) = 0$$
As the authors demonstrate the expectation is never 0.3 when a=1. Therefore r(x,a) is not counterfactually fair.
4. A more minor point compared to the others but I think the authors should slightly tone down what can be guaranteed by Zuo et al (2023). The authors describe this as "the most general way to guarantee counterfactual fairness to our best knowledge". I think firstly that this should be caveated with "given the knowledge of the structural causal model". The main issue with achieving counterfactual fairness is finding the correct structural causal model, not sampling from it given knowledge of the model. Further, given my point in 2 I don't really understand why this is any more general than the fair learn algorithm presented in Kusner 2017, but this seems more to be an issue with Zuo et al (2023) as opposed to this work.

[1] - Counterfactual Invariance to Spurious Correlations: Why and How to Pass Stress Tests. Victor Veitch, Alexander D'Amour, Steve Yadlowsky, Jacob Eisenstein
[2] - Learning Counterfactually Invariant Predictors. Francesco Quinzan, Cecilia Casolo, Krikamol Muandet, Yucen Luo, Niki Kilbertus
[3] - Results on Counterfactual Invariance. Jake Fawkes, Robin J. Evans
[4] - Selection, Ignorability and Challenges With Causal Fairness. Jake Fawkes, Robin Evans, Dino Sejdinovic.
[5] - Counterfactual Fairness Is Basically Demographic Parity. Rosenblatt & Witter (2023)

---

> ### Author Response · Authors · 2025-01-02
>
> * Weakness 1: Thank you for the related work you provided. We added them to the introduction (Section 1) in the revised edition. For the clarification of the difference between our work and the cited work [5], we explained in the response for Weakness 2.
>
> * Weakness 2: You are right about the conclusion that $H(X, A)$ can be written as a function of $U$. As a result, the proof technique presented in Rosenblatt & Witter (2023) can be applied to $H$. However, their proof technique is not applicable to $r(X,A)$ because it is an expectation over U and is not a function of U.
>
> * Weakness 3: We would like to clarify the generation process of counterfactually fair representation. We provide example 1 in section 2.2 in the revised edition and hope it helps.  For this specific example, the first step for generating $r_{A \leftarrow a}(x, a)$ is to infer the conditional distribution of $U$, which is $\Pr(U = -1|X=x,A=a) = 0.4$, $\Pr(U = 0|X=x,A=a) = 0.3$, $\Pr(U = 1|X=x,A=a) = 0.3$. Then we calculate the value of $\frac{x + \check{x}}{2}$ which is $\frac{f(u, a) + f(u, \check{a})}{2}$ ($f$ is defined as Eq. 105) for every $u$. $r_{A \leftarrow a}(x, a)$ is the expectation of $\frac{f(u, a) + f(u, \check{a})}{2}$. To calculate $r_{A \leftarrow \check{a}}(x, a)$, we need to use the same conditional distribution of $U$, then substitute $a$ with $\check{a}$. Therefore, we need to compute $f(u, \check{a}) + f(u, a)$ now. At last, $r(x, a)_{A \leftarrow \check{a}}$ is the expectation of $\frac{f(u, \check{a}) + f(u, a)}{2}$, which is also 0.3.
>
> * Weakness 4: Thank you for your suggestion. We added the expression to the revised edition that the counterfactually fair representation requires the structural causal model to be known. This is also the case in Kusner et al. 2017. We also revised the paper and mentioned that counterfactually fair representation is an extension of (Kusner et al. 2017) and improves performance as stated in Zuo et al. (2023).
>
> * Requested changes 1: As we explained in Weakness 2, we do not claim that $H(X, A)$ cannot be written as $(f(U), U)$. On the other hand, $r(X, A)$ cannot be written as a function of $U$ because of the expectation over $U$.
>
> * Requested changes 2: We explained the question in response to weakness 2 and 3.
>
> * Requested changes 3: We added the discussion about our work and counterfactual invariance work in the introduction in the revised edition.
>
> * Requested changes 4: Thank you for your suggestion. We will emphasize the assumptions that the structural causal model must be given in the revised edition.

---

> ### Comment · Reviewer_dZHZ · 2025-01-08
> **Concerns on the conclusion involving r(X,A)**
>
> I thank the authors for their changes and responses to my concern. Whilst it was not my suggestion I also feel the improved clarity in section 2.2 makes the paper much easier to follow.
>
> My main remaining concern surrounds the r(X,A) representation. So far if I were to summarise the results claimed by the authors on this we have:
> 1. r(X,A) is a counterfactually fair representation.
> 2. r(X,A) need not satisfy parity, even when $U\perp A$.
>
> Together this would lead to the conclusion that counterfactually fair predictors need not satisfy demographic parity in the case that $U\perp A$. This statement is not correct for exact counterfactual fairness. For a proof of this, suppose we satisfy counterfactual fairness, so that:
>
> $$P(r(X,A) = r | X=x,A=a ) = P(r(X(a^{\prime}),a^{\prime}) = r | X=x,A=a )$$
>
> Now, as $U\perp A$ and $X(a^{\prime}) = f(U,a^{\prime})$ we have that $X(a^{\prime}) \perp A$ and so $r(X(a^{\prime}),a^{\prime}) \perp A$. We can put this together to give:
> \begin{align}
> P(r(X,A) = r | A=a^{\prime} ) &= P(r(X(a^{\prime}),a^{\prime}) = r | A=a^{\prime} ) \text{ ( Consistency Property) } \\\\
> &= P(r(X(a^{\prime}),a^{\prime}) = r | A=a ) \text{ (as $r(X(a^{\prime}),a^{\prime}) \perp A$)} \\\\
> &= \mathbb{E}_{P(X|A=a)} [ P(r(X(a^{\prime}),a^{\prime}) = r | X=x,A=a ) ] \text{ ( Law of total expectation)} \\\\
> &= \mathbb{E} _{P(X|A=a)} [ P(r(X,A) = r  | X=x,A=a ) ] \text{ ( Counterfactual fairness constraint)} \\\\
> &= P(r(X,A) = r  | A=a ) ] \text{ ( Law of total expectation)}
> \end{align}
> Therefore, as $P(r(X,A) = r | A=a^{\prime} ) = P(r(X,A) = r  | A=a )$ we have $r(X,A) \perp A$ if $r(X,A)$ satisfies counterfactual fairness. The example the authors give in Appendix D where r(X,A) does not satisfy parity is therefore also sufficient to show that r(X,A) does not satisfy counterfactual fairness in general.
>
> Checking the original paper of Zuo et al on counterfactually fair representations, I can't obviously find a mention of r(X,A) and so I can't find any justification as to why such a representation should be counterfactually fair. So to summarise my remaining concerns are surrounding the correctness of the following two statements claimed by the authors:
>
> 1. r(X,A) has to be a counterfactually fair representation.
> 2. Counterfactually fair predictors need not satisfy parity in the case that $U\perp A$.

---

> ### Author Response · Authors · 2025-01-10
>
> Thank you for your response. In our paper, we defined $r(x, a)$ as
>
> $$r(x, a) = \mathbb{E}_{U \sim \Pr(\mathcal{M})(U|X = x, A = a)}\left[s(\mathbb{E}\left[\check{X}[\check{a}^{[1]}]|U\right], ..., \mathbb{E}\left[\check{X}[\check{a}^{[|\mathcal{A}|]}]|U\right]), U\right] $$.
>
> In our notation, the arguments of $r(\cdot, \cdot)$ are the conditions. Therefore, using notation $r(X(a’),a’)$  does not make sense. Note that in both $ r_{A \leftarrow a}(x, a)$ and $r_{A\leftarrow a'}(x, a)$, the expectation is with respect to $U|X=x,A=a$. Therefore, we also want to respectfully ask the reviewer to take a look at the above formula and verify that $r_{A\leftarrow a’}(x, a) $ and $r_{A\leftarrow a}(x, a)$ are exactly the same.
>
> We want to also emphasize that, the correct notation for counterfactual fairness is,
>
> $$\Pr(r_{A \leftarrow a}(x,a) = r| X=x,A=a) = \Pr(r_{A \leftarrow a'}(x,a) =r| X=x,A=a)$$
> To calculate distribution of $r(X,A)$, we first should note that $r(x,a)$ is a constant vector which is a function of $x$ and $a$. Therefore, distribution of $r(X, A)$ is $\Pr(r(X,A) = r) = \sum_{x,a: r(x,a) = r}p(x, a)$.
> If we want to show that $r(X,A)$ is independent of $A$, we need to show that
>
> $$\Pr(r(X,A) = r|A=a) = \sum_{x: r(x,a) = r} p(x|a)= \sum_{x: r(x,a’) = r} p(x| a’) = \Pr(r(X, A) = r|A = a') $$
>
> For the problem whether $r(x, a)$ satisfies counterfactual fairness, $r(x, a)$ is used in the implementation of Zuo et al. 2023. Do you agree that $H(x, a)$ satisfies counterfactual fairness? If yes, then because $r(x, a)$ is the expectation of $H(x, a)$,  $r(x, a)$ is also counterfactually fair. If you have different opinion about $H(x, a)$, we are glad to answer your questions. Please also elaborate more why the example in Appendix D shows $r(x, a)$ does not satisfy counterfactual fairness. We would be happy to discuss this further.

---

> > ### Comment · Reviewer_dZHZ · 2025-01-13
> >
> > In my previous comment I wrote counterfactuals using the potential outcome notation, in order to be consistent with Pearl's nation used by the authors I will change to $X_{A \leftarrow a}$ for $X(a)$.
> >
> > Firstly and very importantly, the notation the authors give for counterfactual fairness in their response **is not correct**. There are two problems with writing
> >
> > $$P(r(x,a) = r | X=x,A=a ) = P(r_{A \leftarrow a}(x,a) = r | X=x,A=a ).$$
> >
> > Firstly, the notation $r_{A \leftarrow a}$ does not make sense as r is a function from the domains of x and a to $\mathbb{R}^n$ and interventions are defined on variables not functions. In the original paper $\hat{Y}$ is a random variable corresponding to the prediction and so writing $\hat{Y} _ {A \leftarrow a}$ is correct. I feel some confusion on this point stems from the fact that in the original counterfactual fairness paper the authors wrote $\hat{Y}_{A \leftarrow a}(U)$ which is because $\hat{Y}$ can be written as a function of the exogenous variables $U$ via the SCM assumption. However $\hat{Y}$ is still a variable not a function and so we can write $\hat{Y} _ {A \leftarrow a}$. The same argument does not apply to $r$ and so we cannot write $r _ {A \leftarrow a}$ (looking at the draft we shouldn't even write $H _ {A \leftarrow a}$ as again $H$ is a function not a random variable). The role of $\hat{Y}$ and $U$ are discussed in depth in [Silva 2024].
> >
> > Secondly, when intervening for counterfactual fairness it is important that we propagate the intervention through the covariates, which the authors do not. The correct way to write the counterfactual fairness condition for $r$ is:
> >
> > $$P(r(X _{A \leftarrow a}  ,A) = r | X=x,A=a ) = P(r(X _ {A \leftarrow a^{\prime}}, a^{\prime}) = r | X=x,A=a ).$$
> >
> > Which due to the consistency of counterfactuals can be written as:
> >
> > $$P(r(X ,A) = r | X=x,A=a ) = P(r(X_{A \leftarrow a^{\prime}}, a^{\prime}) = r | X=x,A=a ).$$
> >
> > Which is the same as I have written above but now in the Pearl counterfactual notation. This can be seen by looking at [Russell 2017] which is by the same authors where they write counterfactual fairness as:
> > $$P( \hat{Y} _ {A \leftarrow a}= y | X=x,A=a ) = P(   \hat{Y} _ {A \leftarrow a^{\prime}}= y | X=x,A=a ).$$
> > And comment below that $ \hat{Y} _ {A \leftarrow a} = f(x_ {A \leftarrow a},a).$
> >
> > Finally I do not find the response for the authors arguing that $r(X,A)$ is counterfactually fair to be convincing. The authors say that because $H(X,A)$ is counterfactually fair, $r(X,A)$ must be as it is the expectation of $H(X,A)$. This assumes that the space of counterfactually fair predictors is closed under expectation which again **is not the case**. For an easy argument that $r(X,A)$ is not counterfactually fair, note that r is a deterministic function and so there exists some value $r^{\prime}$ such that:
> >
> > $$P(r(x,a) = r^{\prime} | X=x,A=a ) = 1$$
> >
> > On the other hand $P(r(X_{A \leftarrow a^{\prime}}, a^{\prime}) = r^{\prime}  | X=x,A=a ) \neq 1$ in general as $r(X_{A \leftarrow a^{\prime}}, a^{\prime})$ need not be deterministic given $X=x,A=a$. Again Appendix D is a good example of this.
> >
> > Giving the full argument using Appendix D, when $ A=-1,X=0$ we have that $r(X,A) = 0.3$ with probability 1. In terms of counterfactually setting $A=1$ given $A=-1,X=0$, we have that $X_{A \leftarrow 1}$ is 1 with probability 0.6 and 0 with probability 0.4. In the case that $X_{A \leftarrow 1} = 1$ we have $r(X_{A \leftarrow 1}, 1) =0.5$ and if  $X_{A \leftarrow 1} = 0$ we have $r(X_{A \leftarrow a^{\prime}}, a^{\prime}) =0$. Putting this altogether we have that:
> >
> > $$P(r(X,A) = 0.3 | X=0,A=-1 ) = 1$$
> >
> > But
> >
> > $$P(r ( X_{A \leftarrow 1}, 1) = 0.3 | X=0,A=-1 ) = 0$$
> >
> > As $P(r ( X_{A \leftarrow 1}, 1) = 0.5 | X=x,A=a ) = 0.6$ and $P(r ( X_{A \leftarrow 1}, 1) = 0 | X=x,A=a ) = 0.4$. Therefore $P(r(X,A) = 0.3 | X=0,A=-1 ) \neq P(r ( X_{A \leftarrow 1}, 1) = 0.3 | X=0,A=-1 ) $ and so $r$ does not satisfy counterfactual fairness.
> >
> > [Silva 2024] - Counterfactual Fairness Is Not Demographic Parity, and Other Observations. Ricardo Silva.
> >
> > [Russell 2017] - When Worlds Collide: Integrating DifferentCounterfactual Assumptions in Fairness. Chris Russell,Matt J. Kusner, Joshua R. Loftus, Ricardo Silva.

---

> > > ### Author Response · Authors · 2025-01-15
> > >
> > > Thank you for your response. Before we provide a very comprehensive response, we want to make sure that we are on the same page.
> > >
> > >
> > > When we calculate the representation $r$, we do the expectation over the $\Pr_{\mathcal{M}}(U|X = x, A = a)$. In the counterfactual world, where the individual belongs to group $a’$, for calculating $r$, still we use the expectation over $\Pr_{\mathcal{M}}(U|X = x, A = a)$. Our understanding about counterfactual world is that the distribution of U should be calculated given $(x, a)$,  not given $(x_{A\leftarrow a’},a’)$, and we should also calculate the expectation to get our representation using $\Pr_{\mathcal{M}}(U|X = x, A = a)$.
> > >
> > >
> > >
> > > Therefore, in the example that you mentioned, in the counterfactual world, the expectation of $\frac{X + \check{X}}{2}$ is $r = 0.6 * 0.5 + 0 * 0.4 = 0.3$ . Therefore, $\Pr (r(x, a)_{A \leftarrow 1} = 0.3| X=0,A=-1) =1$.

---

### Decision · Action_Editor_ZUBk · 2025-01-15

**Recommendation:** Reject

**Comment:**

The reviewers collectively find that the paper lacks sufficient mathematical rigor, is not clearly positioned within the existing literature to the point where it is hard to judge what are original contributions, treats sensitive attributes inconsistently, employs a non-standard structural causal model without clearly justifying or explaining this choice, and the theoretical results on parity and counterfactual fairness appear to be partially incorrect or established previously. These concerns lead them to conclude that the paper is not ready for publication at TMLR in its current form. I agree with their assessment and recommend rejection, but encourage the authors to take the feedback of the reviewers into account for a future submission with major revisions.

**Audience:**

Yes, the topic and core idea of the paper are relevant to the TMLR audience.

**Claims And Evidence:**

One of the main criticisms of the reviewers is that the claims made in the paper are not properly backed by the theoretical treatment within the paper. Throughout the discussion phase, reviewers were not convinced that the claims made in the paper are correct as stated and still have valid concerns about the rigor and validity of the proofs.

**Resubmission Of Major Revision:**

The authors may consider submitting a major revision at a later time.